# The Intricacies of Renal Phosphate Reabsorption—An Overview

**DOI:** 10.3390/ijms25094684

**Published:** 2024-04-25

**Authors:** Valerie Walker

**Affiliations:** Department of Clinical Biochemistry, University Hospital Southampton NHS Foundation Trust, Southampton General Hospital, Southampton S016 6YD, UK; valerie.walker@uhs.nhs.uk; Tel.: +44-2381206436

**Keywords:** regulator of G protein signalling 14 (RGS14), fibroblast growth factor 23 (FGF23), parathyroid hormone/parathyroid hormone related protein receptor (PTH/PTHrP receptor), Na^+^/H^+^ exchange regulatory cofactor-1 (NHERF1), NaPi-2a sodium-dependent phosphate transporter-2a (NaPi-2a), kidney stones, tumour-induced osteomalacia (TIO), hyperphosphatemia, hypophosphatemia, allosteric regulation

## Abstract

To maintain an optimal body content of phosphorus throughout postnatal life, variable phosphate absorption from food must be finely matched with urinary excretion. This amazing feat is accomplished through synchronised phosphate transport by myriads of ciliated cells lining the renal proximal tubules. These respond in real time to changes in phosphate and composition of the renal filtrate and to hormonal instructions. How they do this has stimulated decades of research. New analytical techniques, coupled with incredible advances in computer technology, have opened new avenues for investigation at a sub-cellular level. There has been a surge of research into different aspects of the process. These have verified long-held beliefs and are also dramatically extending our vision of the intense, integrated, intracellular activity which mediates phosphate absorption. Already, some have indicated new approaches for pharmacological intervention to regulate phosphate in common conditions, including chronic renal failure and osteoporosis, as well as rare inherited biochemical disorders. It is a rapidly evolving field. The aim here is to provide an overview of our current knowledge, to show where it is leading, and where there are uncertainties. Hopefully, this will raise questions and stimulate new ideas for further research.

## 1. Introduction

Phosphate is essential for bone synthesis and turnover and numerous cellular processes including biosynthesis of structural molecules, metabolism, generation of ATP, and intracellular signalling. In healthy adults, almost 80% of total body phosphate is in the skeleton, and only around 0.1% in the extracellular fluid [1,2,3,4,5]. The body content is maintained by balancing phosphate absorption from food with removal by renal excretion. A balanced diet supplies approximately 1 g of phosphorus per day, but more with high meat or dairy consumption. Trans-cellular absorption from the intestine is facilitated by 1,25[OH]_2_D (1,25-dihydoxyvitamin D), but on a normal diet, most influx is via the paracellular route. As this is not tightly regulated, absorption increases with phosphate consumption [6,7,8]. Fine regulation of body phosphorus is entirely dependent on the kidneys, instructed by circulating hormones (notably, parathyroid hormone (PTH), fibroblast growth factor 23 (FGF23), and dopamine), insulin-like growth factor 1 (IGF1), phosphate load, and other agents [8,9,10,11,12,13]. This is a daunting task. The human kidney has around 1.3 × 10^6^ functioning units [nephrons]. Each day, 160–170 L of plasma ultrafiltrate passes across the glomerulus into the kidney at a rate of approximately 1.5 nL/s per nephron [14]. The ratio of filtered phosphate to plasma phosphate concentration is between 0.89 and 0.96 [1]. On a normal diet, filtered phosphate is roughly 170–180 mmol/24 h. Between 85 and 95% is subsequently reabsorbed.

Clinically, the importance of the renal regulation of body phosphate is only too apparent in chronic renal failure and in rare inherited disorders when the process fails [8,9,12,15,16,17,18,19,20]. A frequently encountered problem is reduced phosphate reabsorption in many individuals who have common calcium kidney stones, which is still unexplained [8,21,22]. Treatment is empiric with a risk of causing calcium phosphate precipitation in the kidneys [9]. We need a better understanding of the cellular mechanisms of phosphate transport to improve management. These have been researched intensively for decades. With the very powerful analytical tools now available, the complexity, speed, and amazing co-ordination of the processes involved have become increasingly apparent [11,12,23,24]. Already, findings have shown new avenues for pharmacological intervention to regulate phosphate in common conditions, including chronic renal failure and osteoporosis, as well as rare inherited biochemical disorders [12,19,23,25,26,27,28,29,30,31,32,33,34].

This review focuses on the cellular mechanisms of phosphate reabsorption in proximal renal tubular cells. The aims are to integrate findings from research of the many aspects of the process, to obtain a cohesive view of our current knowledge, and to highlight areas of uncertainty. A brief overview of the anatomy of the proximal tubular lining and its role in phosphate reabsorption is followed by descriptions of the chief executers, the amino acid transporters and the Na^+^/H^+^ exchange regulatory cofactor-1 (NHERF1). Next, their regulation by hormones and dietary phosphate are described, and then, possible mechanisms of phosphaturic hormones produced by tumours are described.

## 2. The Proximal Renal Tubule

Amazingly, almost all renal phosphate reabsorption occurs in the first segments of the nephrons, the short proximal convoluted tubules (PCTs), which have an average length in humans of only 14 mm [10]. The PCTs have three morphologically distinct segments: S1 (pars convoluta), S2, and S3, which are associated with differences in functional activity. The pars convoluta is lined by columnar cells with long tightly packed microvilli which form a tall brush border, lateral cell processes which interdigitate extensively with adjacent cells, and a prominent endocytic-lysosomal system [35,36]. Each microvillus has a central actin core, scaffolding or PDZ (PSD-95/Discs-large/ZO1 domain) proteins, to link membrane proteins to the actin cytoskeleton and myosin VI [37]. This segment absorbs most of the phosphate and much of the fluid from the glomerular filtrate [38,39] and transports H^+^ ions into the tubular lumen via the sodium–hydrogen exchanger 3 (NHE3) transporter. In rats, the pH falls along the proximal tubule from 7.25 in the glomerular filtrate to approximately 6.70 [40]. The villi are shorter in S2, and endocytic vacuoles are smaller; the structure of S3 is simpler [35,36]. Recent investigations have found variation in transcriptomes [41] and metabolic autofluorescence signals along the PCT [42].

## 3. Executors of Phosphate Reabsorption: Renal Phosphate Transporters and NHERF1 (Na^+^/H^+^ Exchange Regulatory Cofactor 1)

Two essential executors of the phosphate absorption process are a responsive transport system to transfer phosphate from the renal tubular lumen into the cells and an intracellular systems operator, NHERF1, to co-ordinate phosphate uptake and the cellular response.

### 3.1. The Renal Phosphate Transporters

Phosphate reabsorption is mediated by at least three sodium-dependent phosphate transporters belonging to solute carrier families SLC34 and SLC20, NaPi-2a, sodium-dependent phosphate transporter-2a (NPT2A, Npt2a, SLC34A1), NaPi-2c sodium-dependent phosphate transporter-2c (NaPi-2c, SLC34A3), and sodium-dependent phosphate transporter PiT-2 (Pit-2, Npt3, Ram-1, SLC20A2), sited in the apical brush border membrane (BBM) [12,24,43], Figure 1. They actively transport phosphate into the cells. The driving force is the inwardly directed electrochemical gradient of Na^+^ ions established by Na^+^/K^+^-ATPase located in the cell basal membranes. There is uncertainty about which proteins transport phosphate out of the cells across the basolateral membrane [4,15,24,44,45,46,47,48]. mRNA abundance of *Slc34a1* is about ten times higher than *Slc34a3* in mouse kidney [49], and in mouse and rat kidneys is >50-fold higher than *slc20a2* mRNA [50]. Immunoreactive NaPi-2a has been demonstrated in the BBM and subapical vesicles of the S1 section of rat proximal tubules and decreases gradually along the straight S2 section [43].

#### 3.1.1. NaPi-2a

NaPi-2a accounts for around 70–80% of overall renal phosphate reabsorption [47]. The gene was identified and cloned in 1993 [24,51]. Human NaPi-2a [Uniprot Q06495] has 639 amino acids. It is predicted to contain two sets of transmembrane-spanning domains separated by a large extracellular loop and cytoplasmic carboxyl and amino terminals. The first intracellular and third extracellular loops may be part of a permeation pore [12,52,53,54,55]. The three carboxyl-terminal residues comprise a PDZ binding domain which can bind the carrier to proteins containing sequences of 80–90 amino acid residues that typify PDZ domains [44,55,56]. NaPi-2a binds to the PDZ scaffolding protein NHERF-1 (Na^+^/H^+^ exchange regulatory cofactor 1) and to other cytosolic scaffolding proteins (Section 3.2). HPO_4_^2−^ is the preferred substrate. Transport is inhibited by PTH, FGF23, and dopamine (Section 4, Section 5 and Section 6).

NaPi-2a proteins probably assemble in the apical membrane as dimers, but the two protomers function independently [12,24]. The rate of phosphate reabsorption is proportional to the number of active transporters on the cell surface and is not dependent on functional changes in the transporter itself [24]. Availability of Na^+^ and phosphate, pH, and membrane voltage influence transport rate. Phosphate affinity is voltage-dependent. At neutral pH and with 1 mM phosphate, the affinity for phosphate is 50–100 μM and for Na^+^ around 50 mM [44,53,57,58]. pH modulates transport by altering the HPO_4_^2−^/H_2_PO_4_^−^ ratio, by reducing sodium affinity at one or more binding sites, and by directing the orientation of the empty carrier between inward- and outward-facing. Reducing extracellular pH over the range 8.0–6.2 reduces Na-dependent phosphate uptake significantly (up to 80%) [44,59]. In the proximal tubule, the pH can fall from approximately 7.4 in the glomerular filtrate to around 6.6, decreasing the ratio of HPO4^2−^/H_2_PO_4_^−^ from 4.0 to 1.6 [50]. Phosphate is transported with a 3:1 ratio of Na^+^ to HPO_4_^2−^, and hence, each transport cycle carries net positive charge across the membrane [12,24,44,52,53,57,58]. In a proposed model, phosphate transport by NaPi-2a is viewed as a kinetic cyclic process comprising a sequence of transitions between conformationally distinct states of the protein [44,52,60].

##### NaPi-2a Partitioning into Lipid Rafts

Around 80% of apical NaPi-2a is localised in lipid rafts in the plasma membrane, which are domains enriched in cholesterol, sphingomyelin, and GM1 glycosphingolipids [12,61]. In vitro, an increase in the cholesterol content decreases NaPi-2a activity of the BBM and is associated with a parallel reduction in membrane fluidity [62]. This situation is observed in ageing rats who have impaired renal tubular phosphate transport, which is not fully explained by reduced *NaPi-2a* mRNA and protein abundance [63,64]. Cholesterol depletion reverses these abnormalities in renal cell cultures and BBM. In K^+^ deficiency, NaPi-2a partitions in rafts enriched with sphingomyelin, glucosylceramide, and ganglioside GM3. Reduced fluidity of the BBM and lateral mobility of NaPi-2a are associated with decreased transport activity. The abundance of NaPi-2a in the BBM is increased [65,66], and NaPi-2a forms pentamers rather than dimers [61]. Normally, within minutes after PTH stimulation of the apical PTH receptor, NaPi-2a moves laterally in the BBM before being endocytosed for destruction in lysosomes [67]. Perhaps an increase in membrane lipids hinders the turnover of membrane-bound NaPi-2a. An increase in BBM cholesterol may partially account for the observation that tubular phosphate reabsorption was significantly lower in men over 50 y attending a renal stone clinic than in younger men [21].

#### 3.1.2. NaPi-2c

*NaPi-2c* was first cloned from human and rat kidney in 2002 [68] and mouse kidney in 2003 [69]. It is found exclusively in mammals [59]. In mice, it is predicted to mediate 15–30% of phosphate reabsorption [3,70,71], but it may have a more important role in humans [47,72,73]. The affinities for Na^+^ and phosphate are 30–50 mM and 0.07 mM, respectively [68,74]. Transport is electroneutral with 2:1 Na^+^:HPO_4_^2−^ stoichiometry. NaPi-2c is very sensitive to pH, but transport is insensitive to membrane voltage [53]. Acidosis reduces mRNA expression, but not membrane abundance, and probably has a direct inhibitory effect on phosphate transport [75]. Although NaPi-2c lacks a canonical C-terminal PDZ-binding domain, it binds to the scaffolding proteins NHERF1 and NHERF3 (Na^+^/H^+^ exchange regulatory cofactor-3, alias PDZK1). In contrast to NaPi-2a, NaPi-2c has a much stronger affinity for NHERF3 than for NHERF1. This is thought to contribute to its strong tethering to the apical membranes of the proximal tubular microvilli, where NaPi-2c and NHERF3 are co-expressed [76,77]. NaPi-2c is internalized through a microtubule-dependent pathway which requires myosin VI [76,77]. Unlike NaPi-2a, it is not degraded but accumulates in a subapical compartment, possibly including recycling endosomes, but this requires proof [77].

High and low phosphate intakes, respectively, decrease and increase activity but at a much slower rate than observed for NaPi-2a [47,77,78]. Potassium deficiency decreases mRNA expression and, in addition, sequesters the transporter in intracellular vesicles, resulting in reduced apical expression and phosphaturia [65]. NaPi-2c transport is inhibited by PTH and FGF23 [47,77,79,80]. The response to PTH is very slow and occurs over several hours in rodents [47,77]. In opossum kidney (OK) cells, siRNA inactivation of *NaPi-2c* significantly suppresses the expression of NaPi-2a protein and mRNA, suggesting that NaPi-2c is important for the expression of NaPi-2a [81]. Expression of NaPi-2c is increased in NaPi-2a deficient mice [75,80]. Bi-allelic loss-of-function mutations in *SLC34A3* cause hypophosphatemic rickets with hypercalciuria (HHRH) [82], a disorder with renal Pi wasting, rickets, and kidney stones. An unusual family with apparent HHRH had digenic inheritance of dominant heterozygous mutations in *SLC34A1* and *SLC34A3.* Individuals with both mutations had significantly more severe disease than those with only one of the mutations, suggesting a gene dosage effect [73].

#### 3.1.3. PiT-2

SLC20A1 (PiT-1) and SLC20A2 (PiT-2) are Type 3 sodium-dependent phosphate symporters. Both are expressed in kidneys, but only PiT-2 is localized at the apical membrane of proximal tubular epithelia [50], where it closely overlaps NaPi-2c [44,45,50]. PiT-2 transports two Na^+^ ions for each phosphate. The affinities for Na^+^ and phosphate are ~50 mM and 100 µM, respectively [10,74]. Although transport is electrogenic, the preferred phosphate species is monovalent H_2_PO_4_^−^. The rate of phosphate transport is affected by external pH because this affects the ratio of H_2_PO_4_^−^ to HPO_4_^2−^. The apparent affinity for phosphate is lowest in the pH range 6.2–6.8 but fourfold higher at pH 5.0. PiT-2 may contribute only moderately (~5%) to renal transepithelial Pi transport but is upregulated in *NaPi-2a*^−/−^ mice during metabolic acidosis and may have a compensatory role [75]. High and low phosphate intakes [50], potassium deficiency [65], and PTH [83] regulate mRNA expression and the abundance of PiT-2 in the brush border of renal proximal tubules.

### 3.2. NHERF1—The Systems Manager

#### NHERF1

NHERF1 was first identified in 1993 and cloned from rabbit kidneys in 1998 [84,85]. Others found it as an ezrin-binding partner and named it an ezrin-binding phosphoprotein of 50 kDa (EBP50) [86]. NHERF1 (SLC9A3 Regulator 1) is from a family of cytoplasmic scaffolding proteins. Their main role is to assemble membrane-associated proteins and signalling molecules, kinases, phosphatases, and trafficking proteins transiently in complexes to direct cell signalling or transport activities [11,86]. They are concentrated at the apical surface of polarized epithelial cells. Apical targeting requires microtubule function [86]. Four members of the NHERF family are expressed in the proximal tubules: NHERF1 which is predominant and is expressed at high levels, NHERF2 (SLC9A3 Regulator 2) which has very weak expression, and NHERF3 (NaPi-Cap1, PDZK1) and NHERF4 (NaPi-Cap2, PDZK2) whose roles in phosphate reabsorption are of uncertain significance [56,87,88]. Figure 2 shows NaPi-2a in the apical brush border membrane bound to NHERF1 which, in turn, is tethered to the actin cytoskeleton.

NHERF1 is a multifunctional protein which scaffolds membrane-bound proteins to the sub-apical actin cytoskeleton. This stabilises them at the cell surface and promotes their incorporation into intracellular signalling complexes [11,89,90]. Amongst numerous membrane proteins bound by NHERF1 are NaPi-2a and NaPi-2c, CFTR (cystic fibrosis transmembrane conductance regulator), sodium/hydrogen exchange factor3 (NHE3), PTH1R (the receptor for PTH and PTH-related peptide, PTHrP), and β2-AR (the β2 adrenergic receptor). Cytosolic proteins bound by NHERF1 include RGS14 (regulator of G protein signalling 14) and the phosphokinases PKCα and G protein-coupled receptor kinase 6 (GRK6A) [47,55,56,91,92,93]. It is estimated that 35–50% of apical membrane NaPi-2a is bound to NHERF1 [94]. However, the concentration of NHERF1 greatly exceeds the total amount of membrane-bound NaPi-2a and GRK6A [95]. NHERF1 null mice have hypophosphatemia, increased renal phosphate excretion, and decreased NaPi-2a in apical membranes [94]. Humans with loss-of-function *NHERF1* mutations similarly have hyperphosphaturia and low plasma phosphate [96].

##### Structure

Human and rabbit NHERF1 have 358 amino acids and share 84% identity over the entire length [85,89]. NHERF1 has five features which are essential for its versatile functioning (Figure 3).

First, NHERF1 has two PDZ domains, PDZ1 and PDZ2. These are binding sites for target proteins. Second, it has a C-terminal ezrin-binding domain (EBD). Through binding to ezrin, NHERF1 is tethered to the subapical actin cytoskeleton [86]. Binding of NHERF1 to ezrin allosterically increases the affinity of the PDZ domains for ligands [97,98]. Third, it is a phosphoprotein with 31 ser and 9 thr residues, of which 17 are in the linker connecting PDZ2 to the EBD. Phosphorylation/dephosphorylation of selected residues changes the conformation of NHERF1 and impacts its interaction with bound proteins [11,89,93,94,99]. Fourth, almost 30% of the protein chain is intrinsically disordered, and hence, NHERF1 is flexible. Altered phosphorylation leads to conformational changes in NHERF1 which may cause allosteric disturbances in the distant PDZ2 and C-terminal domains [11,89]. Fifth, it has a PDZ-binding motif (FSNL) at its C-terminal. In the absence of a regulatory hormone, the C-terminal folds back to bind loosely with PDZ2, so forming a closed molecular structure and blocking the access of binding proteins to PDZ2 and the PDZ2-EBD linker. The C-terminal/PDZ2 link is broken by hormone-induced conformational changes, thus releasing inhibition [Graphical Abstract]. Purified NHERF-1 forms a dimer and heterodimers with NHERF2 and PDZK1, which could form an extended NHERF scaffold, but this is still speculative [86].

### 3.3. Hormonal Regulation of Phosphate Transport by NaPi-2a

Four hormones regulate renal phosphate transport by NaPi-2a. It is decreased by PTH/PTHrP, (PTH related protein) dopamine, and FGF23, which promote phosphaturia, and is increased by growth hormone, acting via IGF1, which promotes phosphate reabsorption. The three phosphaturic hormones, their receptors, and signalling are described individually. The mechanisms by which they decrease phosphate transport by NaPt-2a are then considered collectively.

## 4. Regulation of NaPi-2a by Parathyroid Hormone (PTH) and Parathyroid-Hormone-Related Protein PTHrP

### 4.1. PTH

PTH is synthesized in the parathyroid glands as a precursor, pre-pro-parathyroid hormone (pre-pro-PTH). The 25-residue ‘pre’ signal sequence and 6-residue ‘pro’ sequences are cleaved off sequentially, leaving mature PTH (1–84). This is concentrated in secretory vesicles and granules and released into circulation, together with inactive carboxy-terminal fragments cleaved by proteases during storage. PTH (1–84) is cleared rapidly from the circulation and has a plasma half-life of approximately 2 min. Approximately 70% is extensively degraded in the liver. Around 20% of intact PTH (1–84) is filtered by the renal glomerulus. Most is then bound by megalin in the proximal tubular membrane, internalized, and degraded [9]. Carboxy-terminal fragments are also cleared by glomerular filtration. Because their half-life is longer, their plasma concentrations are several-fold higher than those of PTH (1–84). Regulation of PTH secretion has been investigated and reviewed extensively [8,9]. In brief, acute (minute to minute) regulation is by the calcium sensing receptor [CaSR] in response to circulating ionised Ca^2+^ [2,8,100,101,102]. Longer term regulation is by altering transcription of the PTH gene. This is increased by raised plasma phosphate concentrations acting directly [9,102,103] or indirectly by lowering blood calcium and 1,25 dihydroxy vitamin D (1,25[OH]_2_D) levels or by increasing cell proliferation [102]. PTH transcription and expression are suppressed by 1,25[OH]_2_D [102,104] and isolated hypophosphatemia, which also decreases cell proliferation [102]. FGF23 activation of Fibroblast growth factor receptor 1c (FGFR1c) receptor 1 with its co-receptor, α-klotho inhibits PTH production [94,100,105,106] and, in contrast to the kidneys, increases *CYP27B1* (gene for 25-Hydroxyvitamin D(3)1-alpha-hydroxylase) expression in the parathyroid glands [107]. α-Klotho may also have a direct action on the parathyroid glands to stimulate PTH secretion [9,100].

### 4.2. PTHrP

PTHrP is a secreted neuroendocrine peptide which is widely expressed in normal human tissues, and by some tumours, and is essential for foetal development and bone formation. It acts mainly as an autocrine or paracrine hormone to regulate cell proliferation and differentiation and epithelial calcium ion transport [108]. Canonical human PTHrP [Uniprot P12272] has 177 amino acids, with a signal pro-peptide (residues 1–24). There are three principal secretory forms, PTHrP [1–36], PTHrP [38–94], and PTHrP 107–139 (osteostatin), produced by endoproteolytic cleavage and splice variants [108,109]. Further protease cleavage generates multiple protein fragments, some of which may have endocrine activity when released into circulation. The amino terminal of human PTHrP shares close homology with human PTH, with 8 of the first 13 residues being identical, and binds the peptide to the PTH receptor [110]. Hence, PTHrP can simulate most of the actions of PTH, including proximal tubular phosphate transport. The peptide sequence from residue 14 has little similarity to PTH.

### 4.3. The PTH/PTHrP Receptor (PTH1R)

#### 4.3.1. Structure

PTH1R is a class B1 (secretin) G protein coupled receptor [111,112]. It is expressed at high levels in kidney and in osteoblasts of bone and in a wide variety of tissues where it may be a target for PTHrP rather than for PTH [9,113]. PTH1R receptors are expressed in both apical and basolateral membranes of renal proximal tubules and are proposed to bind filtered and circulating PTH, respectively [94,114,115].

Class B1 G protein-coupled receptors (GPCRs) have a large extracellular domain (ECD), also named the N-terminal domain, which is required for ligand binding [23,111,112,116,117]. This has ~120 residues and a conserved fold composed of an N-terminal α-helix and two pairs of antiparallel β-sheets flanked by a long and a short a-helical segment stabilized by three disulfide bridges. N-linked glycosylation sites in the ECD regulate receptor trafficking and ligand binding [117,118,119]. The heptahelical transmembrane domain (TMD) section has ~260 residues and the intracellular C-terminal ~70 residues [23,116,117]. The transmembrane helices are linked by three extracellular loops and three intracellular loops [23,120]. Crystal structures are reported for the N-terminal [111,121,122] but are not achievable for full-length PTH1R. Single particle cryo-electron microscopy (cryo-EM) has revolutionized membrane protein structural biology, with a resolution below 2 Å achievable for rigid protein sequences. However, structures of intrinsically disordered sequences cannot generally be resolved [117,123]. An engineered peptide, long-acting PTH analogue (LA-PTH), with a high affinity for PTH1R was used to solve cryo-EM structures of PTH1R in complex with G protein Gs (G_S_) [23].

#### 4.3.2. PTH/PTHrP Binding

Most binding and signalling studies have used N-terminal 1–34 PTH and PTHrP fragments and truncated hPTH analogues. The N-terminal portion, especially the most N-terminal residues, is critical for PTH1R signalling. Deletion of PTH Ser1 and Val2 significantly reduces cAMP production. The C-terminal amino acids of the fragments (residues 15–34) bind to the PTH1R ECD and have a critical role in receptor selectivity and affinity [23]. A two-step activation model has been proposed [23,116,117,124]. Regions near the C-termini of the hormone ligands interact with the receptor’s ECD. This directs the amino-terminal portions of the ligands to interact with the receptor transmembrane domain. A change in the conformation of the sixth transmembrane domain (TM6) occurs, resulting in a large outward movement of TM6 associated with a substantial kink in its centre. This exposes an intracellular pocket which enables binding of the Gα subunit via its C-terminal and activation of G-protein signalling [23,117,124]. PTH1R activation is relatively slow. The initial N-terminal binding step has a time constant of ~140 ms, and the following interaction of the ligand with the transmembrane core has a time constant of ~1 s [125,126].

#### 4.3.3. Signalling Pathways and Signalling Bias

In the absence of PTH, the Gα subunit of the G protein heterotrimer is bound to guanosine diphosphate (GDP) and associates with the cytosolic loops of the transmembrane domain. PTH binding to the PTH1R receptor causes a conformational change in the Gα subunit. GDP is replaced by guanosine triphosphate (GTP). The G protein dissociates into Gα-GTP and βγ subunits. GTPases hydrolyse the GTP to GDP, and Gα-GDP reassociates with Gβγ. PTH1R signals primarily via Gs, which stimulates adenylyl cyclase activity but can also couple to Gq/11, which activates phospholipase C (PLC), G_12/13_ which regulates Rho guanine nucleotide exchange factors, and G_i/o_, which inhibits adenylyl cyclase activity and interacts with and signals via β-arrestins [124,127,128,129]. The Gγ subunit can couple to diverse transducer proteins, such as a wide array of G proteins, kinases, and arrestin proteins [130,131]. Figure 4 depicts G-protein signalling pathways activated by PTH/PTH1R binding.

Various N-terminal PTH and PTHrP fragments and truncated PTH (1–34) analogues selectively stimulate Gs, Gq, or G protein-independent signalling pathways. These all have modifications in the N-terminal sequences, which interact directly with the transmembrane domain of the receptor [116,128]. There is evidence that binding of the C-terminal of PTH/PTHrP peptides to the ECD can also bias signalling. The C-terminal tips of the C-PTH-type and C-PTHrP-type 1–34 peptides diverge to opposite sides of the PTH1R ECD [23]. In addition, protease cleavage of the ECD enhances coupling efficacy of the receptor to the Gs pathway, while reducing Gq-coupling at the same time, and results in a signalling bias [116]. An important manifestation of bias introduced at the receptor is that PTH can induce prolonged signalling from intracellular sites, while PTHrP signals exclusively from the cell surface [129] (Section 4).

#### 4.3.4. Termination of Signalling

Within seconds after agonist binding, the G protein-coupled receptor kinase 2 (GRK2) initiates receptor desensitization by preventing G protein coupling [132] and by phosphorylating serine residues in the receptor C terminal, which then engages β-arrestins 1 and 2. This leads to the rotation of the N- and C-terminal domains by 20°, binding of clathrin and the clathrin adaptor assembly polypeptide 2 (AP2), and internalization of the receptor–arrestin complex into endosomes [23,129,133,134,135]. Downstream signalling is terminated, and the G-protein subunits reassociate into the heterotrimeric complex [136]. Within the endosomes, the agonist and receptor are dissociated. The receptor is then targeted to lysosomes and degraded or dephosphorylated and recycled back to the plasma membrane [137,138]. In addition, PTH secretion by the parathyroid glands is suppressed by negative feedback when plasma Ca^2+^ is restored [137]. PTH activation of phospholipase C (PLC) signalling is suppressed by receptor phosphorylation and by phosphorylation of PLCβ3 by PKA [139]. Two additional GPCR regulators may be implicated in the desensitisation of PTH1R.

##### RGS14 (Regulator of G Protein Signalling 14)

*RGS14* is expressed in the proximal and distal renal tubules. RGS proteins control signalling through heterotrimeric G proteins by accelerating the intrinsic GTPase activity of Gα subunits, typically resulting in inhibition of downstream G protein signalling pathways (Section 8).

##### Receptor-Activity-Modifying Proteins (RAMPs)

RAMPS are ubiquitously expressed single-pass trans-membrane proteins that interact with class B1 GPCRs, including PTH1R, and modulate their function. There are three RAMPs in mammals [117,124]. PTH1R-and RAMP2 are highly co-expressed in kidneys, and PTH1R expressed in immortalized human embryonic kidney cells (HEK293) has clear preference for RAMP2. It is unknown whether, or how, RAMP proteins affect PTH1R function. RAMP2 has been observed to shift PTH1R bound to PTH (but not to PTHrP) to a preactivated state. This increases Gs and Gi3 activation kinetics in response to PTH and increases both PTH- and PTHrP-triggered β-arrestin2 recruitment to PTH1R, which parallels the overshoot in Gs activation and increases PTH-mediated Gi3 signalling sensitivity [124].

#### 4.3.5. Endosomal Signalling

Förster resonance energy transfer (FRET)-based studies of ligand–PTH_1_R interaction and cAMP production showed that PTHrP_1–36_/PTH1R binding is fully reversible. cAMP increased transiently and ceased after ligand washout. However, PTH_1–34_, remained bound to PTH1R and cAMP generation continued after ligand washout [23]. This protracted cAMP response deviates from the conventional model of GPCR desensitization (Section 4.3.4). It was found that PTH mediates sustained cAMP signalling from early endosomes following internalization of a PTH1R–Gβγ−β-arrestin complex [140]. PTH1R has two distinct active PTH1R conformations (R^G^ and R^0^). The R^G^ conformation is G protein dependent and is associated with transient cAMP responses from the plasma membrane. It is stabilized by PTH_1–34_ and PTHrP_1–36_ indistinguishably. The R^0^ state is stabilized preferentially by PTH. It is not altered by G protein coupling and maintains cAMP production after the receptor is internalized [131,132,133,134]. In the endosomes, the PTH/ PTH1R, β-arrestin, Gβγ complex (1) promotes the activation cycle of G_s_ from endosomes [140], (2) activates extracellular signal-regulated protein kinase 1/2 (ERK1/2) via β-arrestins and thereby decreases cAMP degradation by phosphodiesterase PDE4 [141], and (3) activates adenylate cyclase type 2 via Gβγ [142]. It was shown that formation of the ternary PTH_1_R–Gβγ−arrestin complex is determined by PTH activation of G_q_. Gβγ released upon G_q_ activation stimulates phosphoinositide 3-kinase β (PI3K_β_) conversion of phosphatidylinositol 4,5-bisphosphate (PI(4,5)P_2_) to PI(3,4,5)P_3_. This promotes β-arrestin recruitment to PTH1R and the formation of the ternary receptor complex [143]. At the acid pH in the endosomes, PTH dissociates from PTH1R–Gβγ–arrestin. Inactive PTH1R is released, assembled with the retromer complex, and trafficked to the Golgi apparatus [141,144] and/or recycled to the plasma membrane. Endosomal cAMP signalling is prolonged by an engineered peptide referred to as LA-PTH, which maintains a higher affinity binding to the PTHR R^0^ conformation than PTH [23,34,142].

#### 4.3.6. Inherited Defects of PTH/PTHrP Receptor Signalling

Observations of patients with naturally occurring mutations of the PTH1R receptor and its associated G proteins have made a valuable contribution to our understanding of PTH/PTHrP signalling [9,145,146,147,148] (Table 1).

### 4.4. Epac1 (Exchange Protein Directly Activated by cAMP 1)

Two intracellular cAMP receptors, protein kinase A, alias cAMP-dependent protein kinase, and exchange protein directly activated by cAMP (Epac), alias cAMP regulated guanine nucleotide exchange factor (cAMP-GEF), mediate the effects of cAMP generated by adenylyl cyclase in response to GPCR stimulation [155,156,157]. Both have a regulatory domain which binds cAMP with equal affinity [158]. By sensing intracellular concentrations of cAMP, this domain acts as a molecular switch to control diverse cell activities. Dependent on intracellular location and prevailing intracellular conditions, PKA and Epac may act synergistically, as in inactivation of NHE3 [159], or antagonistically. Having two cAMP receptors enables differential regulation of generated cAMP, spatially and temporally [155,158]. There are two isoforms of Epac, Epac1 (gene *RAPGEF3)* and Epac2. EPAC1 is the isoform expressed in the kidneys and is detected immunochemically in most of the nephron. In the renal proximal tubules, protein expression is largely restricted to the brush border [160]. Compared with PKA, the role of Epac1 in regulation of proximal tubular function has been largely neglected.

#### 4.4.1. Structure

Epac1 contains an N-terminal regulatory region and a C-terminal catalytic region (Figure 5). The regulatory region comprises a dishevelled/Egl-10/pleckstrin (DEP) domain and the cAMP-binding [CBD] domain. The DEP domain is involved in locating Epac1 at the plasma membrane. The catalytic region consists of a RAS (small GTPase) exchange motif (REM) domain, which stabilizes the active conformation of Epac, a protein interaction motif, the RAS association domain, and a cell division cycle 25-homology domain (CDC25HD), which promotes the exchange of GDP for GTP on Rap (small GTPase) GTPases [155,156,161]. In the absence of cAMP, the CBD associates closely with the catalytic region and prevents activation. cAMP binding to the CBD domain causes a conformational change which removes the autoinhibition of the catalytic domain [156,158].

#### 4.4.2. Actions of Epac1 in the Proximal Tubules

Epac1 is one of the guanine-nucleotide exchange factors (GEFs) which activate the small GTPases, Rap1 and Rap2, in a protein kinase A (PKA)-independent manner [157,162,163,164]. Small GTPases function as molecular switches which regulate signalling events that control numerous cell processes [164]. To date, there are only limited data on the functions of Epac1 in the proximal tubules. Epac signalling has been shown to promote expression and trafficking of the Na^+^-glucose cotransporter type 1 via a mechanism involving caveolin-1 and F-actin in proximal renal tubular cells [159,161]. In the only reported comparison of PKA and Epac on tubular transport, opossum kidney (OK) cells and murine kidney slices were treated with cAMP analogues which selectively activated PKA or Epac. Both PKA and Epac were shown to inhibit NHE3 activity without changing the expression of NHE3 in the BBM. The Epac effect was independent of PKA and, unlike PKA, did not increase NHE3 phosphorylation. However, in contrast to PKA, Epac activation had no effect on NaPi-2a activity and did not induce retrieval of NaPi-2a from the BBM [159]. In an Epac1-deficient mouse model, NHE-3 expression in the proximal tubule decreased by 75%, and the mice had a phenotype consistent with NHE3 deficiency. It was suggested that one action of Epac1 might be to stabilize its interaction with the cytoskeleton at the brush border membrane via a mechanism involving NHERF1 [165].

From the above findings [159], Epac does not have a role in regulating phosphate transport by NaPi-2a. However, OK cells do not express *RGS14*, and RGS14 in rodent cells lacks the C-terminal extension expressed in human renal cells, which was recently reported to bind to NHERF1. RGS14 has a Ras/Rap-binding domain and is a Ras effector [166] (Section 8). It would be interesting to know whether the same results are found in human proximal tubular cells stimulated by PTH.

## 5. Dopamine

Dopamine is a tyrosine derivative made in sympathetic system neurons and in non-neuronal tissues including the kidneys. In the kidneys, dopamine inhibits Na^+^ transport by autocrine/paracrine actions on the sodium-dependent transporters Na^+^/H^+^ exchanger 1 (NHE1), NHE3, NaPi-2a, sodium bicarbonate cotransporter 1 (NBCe1), Na^+^/K^+^-ATPase, and probably the Na^+^Cl^−^ symporter (NCC) [167,168,169]. The major determinants of the renal tubular synthesis/release of dopamine are probably sodium intake and intracellular sodium [167].

### 5.1. Sources and Degradation

The concentration of free dopamine in plasma is low (<1 nmol/L) [167], and dopamine entering the kidneys by glomerular filtration makes little contribution to the urinary excretion, which is in the micromolar range. Up to ~30% of this derives from renal dopaminergic nerves [170,171,172,173,174]. However, most is produced by cells lining the renal tubules by decarboxylation of the dopamine precursor l-dihydroxyphenylalanine (l-DOPA) by aromatic amino acid decarboxylase (AADC) [175,176,177]. This is a circulating intermediate mainly released from sympathetically innervated tissues [178,179]. l-DOPA is transferred from the glomerular filtrate and circulation by SLC7A8 (LAT-2) and SLC7A9/SLC3A1 amino acid transporters in the apical and basolateral membranes [180,181]. The proximal tubules have the highest expression of AADC and contribute most to renal production [167,182]. Dopamine then enters the tubular lumen and activates dopamine 1 receptors (D1R) on the apical membranes in an autocrine/paracrine manner (Figure 6). Paradoxically, in inherited *AADC* deficiency in which brain dopamine is low and circulating l-DOPA is raised, urinary dopamine excretion is normal or high, indicative of residual renal AADC activity [183].

Dopamine is degraded in renal tissues by deamination via monoamine oxidase (MAO), methylation via catechol-O-methyltransferase (COMT) [184,185], and oxidation by renalase. Renalase is a secreted flavin adenine dinucleotide (FAD)-dependent amine oxidase. It is synthesized in the renal glomeruli and proximal tubules and released into the blood and renal tubular lumen [182,186]. Renalase expression and activity are downregulated by an increase in dietary phosphate and might explain why feeding a diet high in phosphate diet increases renal dopamine excretion [172,187,188,189]. Urinary phosphate excretion is increased in renalase knockout mice [190,191].

### 5.2. Dopamine Receptors and Signalling

Dopamine interacts with two families of G protein-coupled membrane receptors, D1 receptors comprising D1R and D5R, and D2 receptors comprising D2R, D3R, and D4R. D1R receptors are highly expressed in the proximal tubules, on both apical and basolateral membranes, and on the renal vasculature, but the subtypes differ [168,192,193]. Dopamine secreted into the tubular lumen acts mainly via D1R in an autocrine/paracrine manner to regulate ion transport in the proximal and distal nephron.

Dopamine binding to the receptor triggers dissociation of the attached trimeric G protein into Gα and Gβγ subunits. Depending on the dopamine receptor subtype, the Gα subunit either activates or inhibits adenylyl cyclase. The βγ subunit recruits G protein-coupled receptor kinases (GRKs) which phosphorylate serine and threonine residues. This promotes binding of arrestins and disrupts the interaction of D1R with G protein subunits [167,194,195,196]. The D1R/β-arrestin complex undergoes endocytosis/internalization via clathrin-coated pits. The dopamine receptors are sorted into recycling endosomes and returned to the cell membrane or transported through the endosome system and degraded in lysosomes. It is likely that D1-like receptors undergo time-dependent desensitization [197]. In humans, D1R and D5R receptors stimulate adenylyl cyclase and activate PKA [198,199,200]. D1R, but not D5R, couples to G_o_ [201]. D1R are also linked to Gαq and activation of PKC [167,202,203,204].

## 6. FGF23 and FGFR1C/KLOTHO Receptor

Unlike the majority of FGFs which bind to cells locally and have confined autocrine/paracrine activities, FGF23 synthesised by bone osteocytes and osteoblasts and by rare mesenchymal tumours enters the circulation and has hormonal actions [205,206]. Low expression has been reported in other tissues but is of unknown significance. FGF23 targets the proximal kidney tubules, where it decreases phosphate reabsorption and reduces 1,25[OH]_2_D production [207], and the distal tubules, where it increases calcium absorption.

### 6.1. Physiological Factors Increasing/Decreasing FGF23 Production

Circulating FGF23 levels are increased physiologically by dietary phosphorus, raised serum phosphorus, 1,25[OH]_2_D, leptin, and other factors [100,208,209,210,211,212]. The activation of FGFR receptors in osteoblasts by locally produced growth factors FGF1 and FGF2 increases FGF23 production [213]. This may be a central pathway for regulating FGF23 expression in bone [8,100,213]. The major systemic regulating factor is 1,25[OH]_2_D which acts via vitamin D receptor (VDR)-dependent and -independent signalling pathways [211]. Dietary phosphate has variable and modest effects on FGF23 in humans [3,100,214]. FGF23 was suppressed in *NaPi-2a* null mice on a low phosphate diet and normalized by a high phosphate intake [215]. PTH increases *FGF23* expression and transcription in vitro and in vivo but also increases furin cleavage, and the effects are variable [100,216,217]. FGF23 is increased in patients with primary hyperparathyroidism or activating mutations of *PTH* or the *PTH receptor* and by continuous PTH infusion, but it is decreased by intermittent PTH infusion to promote bone anabolism and, sometimes, in hypoparathyroidism. A PTH-FGF23 feedback loop has been proposed in which increased PTH increases FGF23 which then decreases PTH [100].

### 6.2. Structure

Nascent canonical FGF23 has 251 amino acids. The first 24 residues are a signalling peptide which is removed [218]. A key feature is a proteolytic cleavage site [R176HTR179], which is hydrolysed by a subtilisin-like proprotein convertase, furin, [207,218,219,220] intracellularly to yield inactive secreted N-terminal and C-terminal fragments with 156 and 71 amino acids, respectively (Figure 7) [218,220,221,222,223]. Extracellular proteases of the plasminogen activation system and plasmin may cleave FGF23 at this site and at other sites [218]. The site is protected from cleavage by glycosylation of Thr_178_ by GALNT3 [*N*-acetylgalactosaminyl transferase 3], whereas proteolysis is promoted by phosphorylation of Ser_180_ by the extracellular kinase family member 20C [FAM20] which inhibits glycosylation [218,219,224].

Individuals with inherited disorders and animal models with increased expression or activity of FGF23 generally have bone abnormalities, hyper-phosphaturia, and low plasma concentrations of 1,25[OH]_2_D and phosphate, unless renal function is impaired. Individuals with inherited disorders causing deficiencies of FGF23 or of its actions have ectopic mineralization of soft tissues including the kidney, impaired renal phosphate excretion, hyperphosphatemia, and hypercalcemia due to high levels of 1,25[OH]_2_D [8,225,226,227] (Table 2). Targeted ablation of FGF23 in mice results in premature death [226].

### 6.3. Renal Receptors and α-Klotho

FGF23 interacts with the FGF receptors FGFR1c, 2c, 3c and fibroblast growth factor receptor 4 (FGFR4) [79]. In the proximal tubules, FGF23 binds preferentially to FGFR1c at the basolateral membrane. The single-pass transmembrane protein α-klotho is an essential partner for binding and signal transduction [218,232,233]. The receptor is a dimer of two units, in each of which the ectodomain of receptor FGFR1c is bound to the ectodomain of the co-receptor α-klotho. This forms a binding pocket for the C-terminal region of c-FGF23 [234]. Heparan sulfate (HS), which has a weak affinity to FGF23 itself, facilitates the formation of a 2:2:2:2 FGF23–FGFR1c–klotho–HS signal transduction unit [234,235]. There is evidence that the circulating free C-terminal fragment of FGF23 produced by furin cleavage competes for binding to the receptor complex and may block the phosphaturic response to intact FGF23 [29,236].

#### α-Klotho

In normal proximal tubules (mice), *α-klotho* is expressed at the mRNA and protein level together with FGFR1, 3, and 4 [237]. In addition to full-length α-klotho, there are two soluble forms: a truncated isoform produced by alternative gene splicing, and the extracellular domain of full-length α-klotho released through cleavage by disintegrin and metalloproteinases 10 and 17 (ADAM 10 and ADAM 17) at the cell surface. Soluble α-klotho is secreted into blood, CSF, and urine and functions as an endocrine/paracrine substance. The kidneys are the main source of circulating soluble α-klotho [100,238]. Intact α-klotho is too large for glomerular filtration [100,238,239].

### 6.4. Signal Transduction and Signalling

Fibroblast growth factors (FGFs) signal through the four FGF tyrosine kinase receptors (FGFR1–4) and thereby activate the RAS-MAPK (mitogen-activated protein kinase) and the phosphatidylinositol 3-kinase (PI3K)/Akt serine/threonine kinase (PI3K-AKT) pathways [240]. The phosphaturic effects of FGF23 in the kidney tubules are klotho-dependent. In the proximal renal tubules, FGF23 reduces the membrane abundance of NaPi-2a and NaPi-2c. The internalization and degradation of these phosphate transporters were shown to depend on the activation of ERK1/2 and serum/glucocorticoid-regulated kinase-1 (SGK1) pathways [237], resulting in phosphorylation of NHERF1 [241]. FGF23 stimulates both fibroblast growth factor receptor substrate 2 (FRS2) and ERK1/2 in the proximal tubule [237,242] (Figure 8). Targeted ablation of proximal tubule klotho increased serum Pi and reduced urine Pi, suggesting that FGF23 regulates re-absorption of Pi in a klotho-dependent manner in the proximal tubules [243]. In the absence of klotho, high concentrations of FGF23 can activate FGFR4. This induces PLC-catalysed production of diacylglycerol and inositol 1,4,5-trisphosphate, which increase cytoplasmic calcium levels [244]. Calcium activation of calcineurin dephosphorylates the transcription factor nuclear factor of activated T-cells (NFAT), which permits its translocation into the nucleus to modulate the expression of specific target genes [245]. The transcription factor EGR1 [early growth response factor 1] is a downstream marker of MAPK signalling [242,246]. In Phex-deficient mice, MAPK/ERK1/2 signalling is activated. A selective mitogen-activated protein kinase (MAPK) kinase (MEK) inhibitor blocks this response [33,247].

## 7. Mechanisms of Hormonal Regulation of Phosphate Transport by NaPt-2a

From a review of studies up to 2012 to investigate the mechanisms of action of the three phosphaturic hormones on NaPi-2a transport, it was proposed that they use a common route which leads to the severance of a link between NaPi-2a and NHERF1 [94]. Binding of NaPi-2a to this scaffolding normally stabilises the transporter in the apical plasma membrane. When the link is broken, detached NaPt-2a is endocytosed and degraded, the number of transporters at the cell surface decreases, and phosphate absorption is reduced. The process is initiated by phosphorylation of NHERF1. The kinases in the PTH and the dopamine pathway are PKA and PKC and probably MAPK in the FGF23 pathway.

The evidence for this proposal was based on the culmination of years of research. (1) Phosphate transport correlates with the amount of NaPi-2a expressed on the apical membranes of proximal renal tubules [94,249,250]. (2) The expression is decreased by clathrin-mediated endocytosis of NaPi-2a and targeting to lysosomes for degradation [67,250,251]. Trafficking of the endocytic vesicles to lysosomes is via microtubules and requires myosin VI and a dynamic actin skeleton [252,253]. (3) Surface expression is decreased by PTH, dopamine and FGF23, and phosphate transport is inhibited within 30–45 min [94,254,255,256,257]. (4) NHERF1-null mice have increased phosphate excretion, low plasma phosphate, and decreased NaPi-2a abundance in the apical membrane [258]. (5) NHERF1 phosphorylation is the leading event in dissociation of NaPi-2a from NHERF1. NaPi-2a binds to the PDZ1 motif in NHERF1. Ser77 is the major phosphate acceptor in the PDZ1 domain for all three hormones. Ser77 phosphorylation decreases NaPi-2a binding to NHERF1 causing dissociation. Thr95 is a secondary acceptor for PTH and dopamine stimulation [259]. However, thr95 phosphorylation blocks the actions of FGF23 on phosphate transport [257]. (6) PTH is expressed both on apical and basolateral membranes of proximal tubular cells. Apically, the PTH receptor binds to NHERF1 PDZ1 and activates PKC [260,261]. However, in the absence of NHERF1 at the basolateral membrane, PTH stimulates cAMP/PKA but still reduces phosphate absorption [262]. In vitro PKC, but not PKA, phosphorylates ser77 directly, suggesting that PKA acts indirectly. (7) At low concentrations, PTH and FGF23 are synergistic, with increased activation of PKC and PKA but not MAPK [257].

Observations from subsequent studies support this proposal. The capacity to probe cell signalling pathways and determine molecular structures and interactions is revealing an amazing dynamic regulatory process. Knowledge of PDZ domains and PDZ-binding partners and their promiscuity has increased [262]. Flexible unstructured protein sequences have gained prominence because of their roles in allosteric regulation of the interaction between domains and in modifying protein molecular conformation and the interaction of amino acid residues [89,263,264].

Using a panel of kinase inhibitors, acute downregulation of NaPi-2a transport by PTH was shown to be mediated by PKA, PKC, and ERK1/2 (MAPK) down-stream of PKC, whereas SGK1 (serum and glucocorticoid-activated kinase) mediated downregulation by FGF23. *NHERF1* knockdown prevented both PTH and FGF23 actions, indicating that the different PTHR and FGFR1c signalling pathways converge at the level of NHERF1 phosphorylation [248]. The PTH receptor PTH1Rwas shown to have a C-terminal PDZ-binding motif [ETVM], which was predicted to interact with both NHERF1 PDZ1 and 2 domains. Determinants outside the PDZ-ligand pocket enhanced formation of the NHERF1-PTH1R complex [93,265,266,267].

In a study of NHERF1 residues phosphorylated by PTH stimulation, Ser290 in the intrinsically disordered PDZ2-ezrin binding domain (PDZ2-EBD) linker displayed a conspicuous transient de-phosphorylation [11]. Ser290 phosphorylation decreased by 90% at 1min after PTH but was fully restored at 5 min. Dephosphorylation was associated with changes in NHERF1 conformation at critical positions for binding NaPi-2a and in the EBD at the C-terminal tail. These were reversed by re-phosphorylation. De-phosphorylation was attributed to a novel NHERF1 binding partner, protein phosphatase 1 (PP1), bound at V257PF259, a putative VxF/W motif. Re-phosphorylation of Ser290 was mediated by GRK6A bound to phosphorylated Ser162 in PDZ2 [11]. Ser162 is known as a PKCα phosphorylation site in human NHERF and is essential for hormone-sensitive phosphate uptake [95]. In the unstimulated state, binding of the NHERF1 C-terminal to PDZ2 would block the access of GRK6.

From these findings, it was proposed that activation of PKC by PTH triggers rapid dynamic de-phosphorylation/re-phosphorylation cycling at Ser290, leading to allosteric conformational changes in NHERF1. These disrupt binding of NaPi-2a to NHERF1, resulting in the internalisation and destruction of NaPi-2a and reduced phosphate absorption [11] (graphical abstract). An interesting observation was that after PTH addition at normal or high extracellular phosphate concentrations in the presence of sodium, there was a transient burst of intracellular phosphate for ~10 min, followed by a decline to resting levels. The authors commented that this was consistent with uptake by NaPt-2a but offered no explanation for NaPi-2a activation. Perhaps one possibility to consider is an increase in extracellular pH [44,45] by PTH inhibition of NHE3.

In addition to its C-terminal TRL PDZ-binding motif, human NaPt-2a has an internal T494RL motif which has not been fully characterised [93]. Two disease-associated mutations in *NHERF1* were identified in this motif, R495H [268] and R495C [269]. Affected individuals had increased phosphate and cAMP excretion and low serum phosphate. Three mutations located in PDZ2 and in the PDZ1-PDZ2 linker were identified in seven patients with decreased phosphate reabsorption. In OK cells, the PTH-stimulated cAMP response was significantly higher in cells transfected with mutant *NHERF1* cDNA than with wild-type cDNA and was associated with a significantly greater decrease in phosphate uptake. There was no difference in intracellular Ca^2+^ concentration or inositol trisphosphate production [96].

FGF23 induces phosphaturia by decreasing the abundance of NaPi-2a in proximal tubular cell membranes through activation of the ERK1/2-SGK1 signalling pathway [96,237,248] (Section 6.4). This action requires NHERF1 and synergizes with PTH [237,257]. It seems likely that FGF23 triggers a dynamic de-phosphorylation/re-phosphorylation episode which causes allosteric conformational changes in NHERF1 and disrupts NaPi-2a/NHERF1 binding as observed for PTH. However, this may not involve Ser290, which is constitutively phosphorylated by GRK6 [11]. There are closely sited serine residues which are alternative potential substrates for phosphorylation by SGK1. The observation that, in contrast to PTH and dopamine, thr95 phosphorylation inhibits the phosphaturic response to FGF23 may indicate a difference between SGK1 and GRK6 in the conformational changes induced by PTH and FGF23, respectively.

## 8. Regulator of G Protein Signalling 14 (RGS14)

Genome-wide association studies [GWASs] to look for common gene variants associated with calcium kidney stones [270,271,272,273,274] and chronic renal failure [275,276] identified significant associations with single intronic nucleotide polymorphisms (SNPs) in the gene encoding one of the Regulator of G protein Signalling proteins, RGS14. These were associated with circulating plasma phosphate concentrations. Recent studies demonstrating that RGS14 binds to NHERF1 in the human renal proximal tubules and prevents inactivation of NaPi-2a by PTH [166] may help to explain this.

RGS proteins control signalling through heterotrimeric G proteins by accelerating the intrinsic GTPase activity of Gα subunits, typically resulting in an inhibition of downstream G protein signalling pathways. RGS proteins are primarily regulated by mechanisms that control their local concentration at the site of signalling. The expression of RGS proteins is highly dynamic and is regulated by epigenetic, transcriptional, and post-translational mechanisms [277]. *RGS14* is expressed in the proximal and distal renal tubules. In human proximal tubular cells expressing protein at endogenous levels, RGS14 co-localises with NHERF1 [166].

### 8.1. Structure

Similar to other RGS proteins, RGS14 has an RGS domain in the N-terminal which binds to Gαi G-protein subunits and blocks G protein signalling. In addition, RGS14 has two flexible tandem Ras/Rap-binding domains, RBD1 and RBD2 (Figure 9). Initial in vitro studies identified the Rap GTPases Rap1 and Rap2 and activated Rap2 as binding partners for the RBD domain, suggesting that RGS14 might be an effector for activated Rap proteins. However, further investigation demonstrated that RGS14 bound selectively to activated H-Ras and not to Rap isoforms [278]. The RGS14/H-Ras association could assemble a multiprotein complex with components of the ERK MAPK pathway [278]. In the RGS14 C-terminal, a G-protein regulator (GPR, alias GoLoco) motif binds to inactive Gαi1/3 to anchor RGS14 at the cell membrane where it has GDI (GDP-dissociation inhibition) activity at the Gαi subunits. This prevents exchange of GDP for GTP and activation of G protein [279,280]. The GPR motif also blocks the association of Gα and Gβγ, potentially leading to prolonged Gβγ signalling [281]. Notably, unlike most other genotyped animals including rodents, RGS14 of humans and primates has a 21-residue extension to the C-terminal which terminates in a PDZ-binding motif (DSAL). This binds to the PDZ2 motif of NHERF1. RGS14 is the only RGS protein that has a canonical PDZ-recognition motif [166].

### 8.2. RGS14 Inhibition of PTH Regulation of NaPi-2a

In human primary kidney cells, *RGS14* knock down unmasked PTH-sensitive Pi transport. However, RGS14 did not affect PTHR-Gs coupling or cAMP production, which indicated an action at a post-receptor site. *RGS14* expression in human renal proximal tubule epithelial cells blocked the effects of PTH and FGF23 and stabilized the NaPi-2a–NHERF1 complex [166]. It was proposed that binding of the C-terminal PDZ ligand of RGS14 to PDZ2 of NHERF1 confers a tonic inhibition of the PTH and FGF23 actions. Against this is the dynamic expression of RGS14 and previous observations in unstimulated hippocampal neurons that RGS14 is most abundant in the cytosol but is recruited to the plasma membrane following stimulation [282]. An alternative scenario might be that cAMP generated by PTH/PTH1R signalling activates Epac and thereby activates Rap1 and/or Rap2 (Section 4.4). Activated Rap might then bind with RGS14 and promote migration of the RGS14 C-terminal to NHERF1 and binding to PDZ2. This would then inhibit the actions of PTH and FGF23 as proposed above. Could Epac/RGS14 interaction be an additional mechanism for regulating phosphate absorption?

### 8.3. RGS14 Polymorphisms in GWAS

Three of the intronic SNPs identified in the GWASs, rs4074995 [275,276], rs56235845 [271], and rs11746443 [272,274], are in the RBD domain of RGS14 (Figure 10). The fourth, rs12654812 [270,273], is in the RGS domain. Because the next down-stream gene from *RGS14* is *SLC34A* which encodes NaPi-2a, it has been thought previously that the *RGS14* SNPs were picking up a mutation which decreases NaPi-2a production or transport activity. The alternative is that they are ‘sensing’ a dysfunctional mutation in *RGS14.* This is an attractive proposition. A mutated RGS14 that cannot effectively block the inhibitory action of PTH on NaPi-2a could produce biochemical disturbances of hyperparathyroidism without increased serum PTH levels. A high proportion of idiopathic calcium stone formers have an increased fractional excretion of phosphate [8,18,21]. In 6% of men attending a stone clinic, this was associated with a low plasma phosphate concentration. PTH was normal [21]. The fact that *RGS14* is under epigenetic control makes it an attractive candidate for a common disorder in which environmental and dietary factors play a large part [8,283].

## 9. Growth Hormone and Insulin-like Growth Factor-1 (IGF-1)

Growth hormone [GH] stimulates phosphate absorption by the proximal renal tubules and reduces phosphate excretion. Plasma phosphate concentrations are increased during growth. They are raised in infants and children, compared with adults [284], and in acromegaly [9]. GH treatment of children with GH deficiency increases plasma phosphate and renal tubular reabsorption of phosphate [285,286]. GH acts on target tissues mainly via stimulation of the insulin-like growth factor-1 (IGF-1), a 7.6 kDa protein synthesised predominantly in liver. In circulation, more than 99% of IGF-1 is bound to IGF-binding proteins (IGFBPs), mostly to IGFBP-3. Some IGF-1 is synthesised in the kidneys, and some derives from the circulation [287].

GH stimulation of phosphate reabsorption in the proximal tubules is mediated by the upregulation of NaPi-2a by IGF-1 bound to high affinity IGFR-1 receptors on the apical and basolateral membranes [288,289,290,291]. Expression and stability of the transporter in the cell membrane were increased, but not NaPi-2a synthesis [290]. IGF-I stimulation of opossum kidney (OK) cells resulted in a dose-dependent increase in tyrosine phosphorylation of the p95 kDa P-subunit of the IGF-I receptor β-subunit. Tyrosine phosphorylation recruits SH2-domain-containing adapter proteins which activate signalling via the phosphoinositide 3-kinase (PI) 3-pathways and the MAPK cascade. However, signalling to these pathways did not appear to be sufficient to induce Pi transport stimulation [291].

## 10. Dietary Phosphate

Dietary phosphate intake varies widely. With high intakes, intestinal absorption is mainly by the paracellular route and is poorly regulated [7,292]. Normally functioning kidneys respond to oscillations in the filtered phosphate load by adjusting the amount reabsorbed.

### 10.1. Acute Changes in Dietary Phosphate

#### 10.1.1. Low Phosphate Intake

Adaptation to a low phosphate diet involves increased insertion of NaPi-2a into the brush border membrane and requires NHERF1. This is followed by increased apical expression of NaPi-2c, and eventually of PiT-2 [50,68,293,294,295]. Whereas phosphate depletion increased apical expression of NaPi-2a in rats and wild type (wt) and vitamin D receptor null mice [2,58,65,70], this response was absent in NHERF1-deficient mice [296,297]. In rats fed a 0.1% Pi diet, the abundance of NaPi-2a, NaPi-2c, and PiT-2 in the kidney was 100% higher than in rats on a 0.6% Pi diet. PiT-1 was not modified. The increase was pH-dependent. In BBM vesicles, adaptation to the 0.1% Pi diet was accompanied by a 65% increase in the V (max) of Pi transport at pH 7.5, compared to the 0.6% Pi diet. At pH 6.0, the increase was only 11%. Metabolic acidosis increased the expression of NaPi-2c and PiT-2 in animals adapted to the low Pi diet. *NaPi-2a* RNA was estimated to contribute 95% to the total mRNA of the Pi transporters, and *NaPi-2a* accounted for 97% of Pi transport at pH 7.5 and 60% of Pi transport at pH 6.0, with little contribution from PiT-2 [298].

#### 10.1.2. High Phosphate Intake

In rats and mice, apical expression of NaPi-2a is rapidly downregulated by a high dietary phosphate intake [3,65,293,299], followed by decreases in NaPi-2c and PiT-2 [50]. This occurs independently of PTH and FGF23 [300]. Acute variations in dietary Pi levels do not alter RNA levels of the transporters [12]. Of several hormones analysed in rats, only PTH seems to be necessary for the early adaptation of renal phosphate transport to high dietary Pi [301]. In healthy humans challenged by acute intragastric or intravenous Pi loading, the increases in phosphaturia were similar and occurred at a similar time post-load. This was preceded by increases in plasma phosphate and PTH. Plasma FGF23 level increased after the onset of phosphaturia [302].

### 10.2. Chronic Changes in Dietary Phosphate

Persistent changes in serum Pi concentration resulting from chronic alterations in dietary phosphorus intake primarily modulate the gene expression of *NaPi-2a* in the proximal tubule [303]. This may be mediated in part by changes in plasma concentrations of hormones including PTH, FGF23, and 1,25[OH]_2_D. In mice, a phosphate-response element has been identified in the *NaPi-2a* gene which binds the mouse transcription factor muE3 (TFE3). Renal *TFE3* expression was upregulated in mice fed a low phosphorus diet [304], suggesting that this activates *NaPi-2a* transcription on a low intake. Physiologically, adaptive hormonal regulation of the abundance of NaPi-2a on the apical membrane proximal tubule may be more important than transcriptional regulation of *NaPi-2a* by TFE3 [305,306,307]. *The activator protein 1 (AP1)*, *nuclear factor erythroid 2-related factor 2 (Nrf2)*, and *early growth response 1 (EGR1)* transcription factors are upregulated in mammalian cell lines in response to increased extracellular phosphate concentration [308,309,310,311]. Increased extracellular Pi, activates the Raf/MEK/ERK pathway and leads to translocation of these transcription factors into the nucleus where they regulate the expression of phosphate-responsive genes.

### 10.3. High Dietary Phosphate and Renal Dopamine Excretion

Feeding a high phosphate diet to rats and mice [188,189] increased the renal excretion of dopamine. In mice, this was attributable to increased dopamine synthesis through upregulation of *AADC* and to significant decreases in renalase and other enzymes which degrade dopamine [94,189,190] (Section 5.1). PKA and PKC were activated, perhaps indicative of dopamine signalling via D1-like receptors. Carbidopa administration to inhibit dopamine synthesis decreased the phosphaturic response to a high-phosphate diet [189]. Collectively, the animal studies indicate that an increase in endogenous dopamine has an important role in acute adaptation to a high dietary phosphate intake [94]. A study of 884 humans similarly found that high dietary phosphate was associated with higher urine dopamine. However, unlike animal models, fractional excretion of phosphate was reduced. This questions the significance of urine dopamine in the acute adaptive response to dietary phosphate in humans [312].

## 11. Tumour-Induced Osteomalacia (TIO)

Tumour-induced osteomalacia (TIO) is a rare paraneoplastic syndrome characterised by renal phosphate wasting and hypophosphatemic osteomalacia. The causative tumours are usually mesenchymal, commonly hemangiopericytomas and giant cell tumours, often benign and frequently difficult to localise [27,313,314,315]. TIO results from over-production of phosphaturic factors. FGF23 is most often the causative agent, but other proteins may be responsible, notably MEPE (matrix extracellular phosphoglycoprotein) and, infrequently, secreted frizzled-related protein 4 (sFRP4) and growth factor 7 (FGF7) [316,317,318,319,320]. In 50 well-characterized phosphaturic mesenchymal tumours, excessive FGF23 production was attributable to a *fibronectin (FN1)–FGFR1 fusion gene* in 21 (42%) tumours and in 3 (6%) to a *FN1-FGF1 fusion gene* [319,321]. Unlike native FGFR1, the chimeric receptors seem to function independently of klotho co-receptor. In one patient with TIO, raised FGF23 production was attributed to a novel NIPBL-BEND2 fusion gene (fused Nipped-B gene and a gene for a protein with two BEN domains). The NIPBL fusion gene promoted cell proliferation possibly via the MYC pathway [322]. Only half of the above 51 tumours had one of these fusion-genes, and other causes remain to be identified.

### 11.1. MEPE

*MEPE* was first cloned and characterised in three tumours causing TIO [323]. Expression of *MEPE* was 10^4^ to 10^5^ times higher in TIO-associated tumours than in control tumours [317]. MEPE is a secreted matrix extracellular phosphoglycoprotein belonging to a family of small integrin-binding ligand proteins, N-linked glycoproteins (SIBLING proteins) [324,325,326]. The predominant isoform expressed in humans has 525 amino acids. It is a flexible N-glycosylated protein with two defined functional domains (Figure 11).

The Ac-100 domain includes an integrin-binding RGD motif and a glycosaminoglycans (SGDG)-binding motif. The ASARM domain is the short carboxy-terminal peptide (21 residues). It is acidic because of its five aspartate and two glutamate residues. It contains eight serine residues, of which five are phosphorylation sites for casein kinase (FAM20C). The phosphorylated ASARM peptide is released from the parent MEPE molecule by extracellular cathepsins at closely located cleavage sites and possibly by PHEX in bone [327].

*MEPE* is expressed in bone osteoblasts and osteocytes and in the brush border membranes of proximal renal tubules [328,329,330]. In bone, MEPE interacts with membrane-bound PHEX (Phosphate Regulating Endopeptidase X-Linked) and DMP1 (Dentin Matrix Acidic Phosphoprotein) to control *FGF23* transcription [326]. In the early stages of osteogenesis, MEPE stimulates bone formation. This requires the AC-100 motif, which probably activates integrin receptors and thereby triggers downstream signalling [324,325]. In the later stages of osteogenesis, cleaved phosphorylated ASARM peptide binds to hydroxyapatite released from osteocytes and inhibits mineralisation [326,330]. Both MEPE and the cleaved ASARM peptide are released from bone into the circulation in normal individuals. Table 3 summarises studies investigating the renal actions of MEPE on phosphate excretion.

Collectively, the findings from the infusion, perfusion, and cell studies show that MEPE increases the fractional excretion of phosphate FEPO4 by decreasing the rate of phosphate transport (V max) by Npt-2a but not the affinity for phosphate. This was explained by decreased expression of Npt-2a protein in the brush border membrane (BBM). Plasma 1,25[OH]_2_D was only measured in one of these studies and was decreased, comparable with FGF23. Over-expression of *mepe* in transgenic mice resulted in extensive renovascular changes which affected sodium balance, with activation of the renin/aldosterone system. The compensatory increase in *NaPi-2a* expression increased phosphate absorption and plasma phosphate was raised, apparently over-riding MEPE activity [330].

Like MEPE, infusion of the ASARM peptide increased FEPO4 and decreased serum phosphate. Plasma 1,25[OH]_2_D increased. In transgenic mice with over-expressed *ASARM peptide* [334], changes in the biochemistry reflected a large increase in FGF23. This, coupled with an increase in PTH, makes it impossible to gauge the contribution of ASARM to hyper-phosphaturia.

#### Mechanism Causing Hyper-Phosphaturia

How does MEPE cause hyper-phosphaturia? From the observed decrease in expression in the BBM, this may be via NHERF1, as for PTH and FGF23 (Section 7). So far, no MEPE receptor has been identified in the kidneys. Circulating MEPE could associate with proximal tubular cells through attachment of the AC-100 domain to integrin(s) and carbohydrates in the basolateral membranes of the proximal tubules. This would then activate signalling via the integrin C-terminal and perhaps could lead to ser290 phosphorylation in NHERF1 by SGRK. This merits investigation. Integrins αvβ3 and αvβ5 are expressed in renal tubules and possible binding partners. The second partner might be a heparan-sulfate containing mucoprotein. A peptide sequence in the syndecan-1 core interacts with αvβ3 and αvβ3 and modulates cell adhesion [337]. An additional/alternative mechanism is the direct inactivation of apical *NaPi-2a* by the ASARM peptide followed by internalisation, as has been reported for the reactive compound phosphonoformic acid [338]. The peptide could be released by proteolytic cleavage of MEPE in the bone or circulation, or even from MEPE synthesised in the renal tubules. It is unknown whether the highly charged ASARM peptide is filtered at the glomerulus.

### 11.2. Secreted Frizzled-Related Protein 4 (sFRP4)

*sFRP4* was cloned from tumours associated with TIO [320]. In rats, sFRP4 infusion increased urine phosphate and FEPO4 and reduced serum phosphate by PTH-independent mechanisms. Renal *NaPi-2a* mRNA levels were unchanged; renal beta-catenin concentration was reduced, and phosphorylated beta-catenin concentration increased. sFRP4 inhibited sodium-dependent transport of cultured OK cells [339]. However, in a study of hemangiopericytomas and various control cell lines, fibroblast growth factor 7 *(FGF7)* and *sFRP-4* were widely expressed in all studied cell lines and tissues and were not tumour-specific [340]. sFrp4 modulates extracellular signalling by soluble secreted Wingless integrated (Wnt) glycoproteins. *Wnts* bind to Frizzled (FZ) class F GPCRs that mediate Wnt signalling, which has wide-ranging functions [341,342,343,344,345,346]. After binding to the receptor, the signal is transduced to the cytoplasmic adapter phosphoprotein dishevelled (Dsh/Dvl), which transiently recruits signalling complexes. Dsh directs the Wnt signal into at least three major cascades, canonical and non-canonical planar cell polarity (PCP) and calcium pathways [341,342,345,347,348,349]. Canonical signalling is mediated by β-catenin and activates T-cell factor/lymphoid enhancer factor (TCF/LEF) transcription factors to regulate the expression of target genes. The non-canonical pathways act independently of β-catenin. The Wnt/Ca^2+^ pathway activates phospholipase C and increases intracellular Ca^2+^, which in turn activates CamKII (Ca^2+^/calmodulin-dependent protein kinase II) and PKC and the nuclear transcription factors Nuclear Factor kappa B (NFkB), cAMP-response element-binding protein (CREB), and NFAT [342,347,350].

sFRP4 has a cysteine-rich domain in the N-terminus, which is 30–40% identical to the Wnt ligand-binding domain of the frizzled receptors [344,351,352]. It binds secreted Wnts and act as a Wnt decoy receptor. This reduces signalling by both canonical and noncanonical Wnt signalling pathways [320,339,343,345,348,349]. In the rat studies, sFRP4 suppressed canonical wnt signalling in vivo [339] and by inference TCF/LEF transcription. There is no clear direct connection between this and the decreased NaPi-2a transport observed, although secondary disturbances in cell signalling might have been responsible. There was no increase in PTH or cAMP to incriminate PKA activation. An alternative is that PKC was activated by non-canonical Wnt/Ca^2+^ signalling. This could then phosphorylate NHERF1 and inactivate NaPi-2a as described for PTH and dopamine (Section 7). However, sFRP4 would be predicted to decrease Wnt/Ca^2+^ signalling. It may be that sFRP4 has a weak affinity for non-canonical Wnts and does not displace them from their receptors and interrupt their signalling.

### 11.3. Fibroblast Growth Factor 7 (FGF7)

FGF7 (keratinocyte growth factor KGF) was identified in tumours associated with TIO [353], and FGF7 and FGF23 were increased in blood draining the tumour site of a patient with TIO [312]. FGF7 inhibited phosphate uptake by renal tubular epithelial cells in vitro [353] and increased phosphate excretion in rats [354]. However, *FGF7* was expressed in a range of cell lines not associated with TIO [340]. In a cross-sectional study of patients with chronic renal failure there was no significant correlation between serum immunoreactive iFGF7 and phosphate, iFGF23, PTH, or 1,25[OH]_2_D [355]. *FGF7* expression was upregulated in cysts from kidneys of patients with autosomal dominant polycystic kidney disease. In vitro FGF7 stimulated the proliferation of cyst-lining epithelial cell by regulating the expression of *cyclin D1* and *P21* genes [356]. In summary, there is experimental evidence that FGF7 causes hyper-phosphaturia, but its association with TIO tumours may be as a mediator of tumour growth rather than as a phosphotonin.

FGF7 is a paracrine-acting FGF which activates the “b” isoforms of FGFR1, FGFR1b, and FGFR2b and contributes to organ development [357]. The FGF receptors (FGFRs) are single-pass transmembrane tyrosine kinase receptors with an intracellular tyrosine kinase domain. Heparan sulphate glycosaminoglycans (HSGAGs) expressed on the cell surface are essential co-receptors. They promote the formation of a symmetric 2:2 FGF: FGFR dimer on the cell surface and activation of the receptors [357]. The mechanism by which FGF7 causes hyper-phosphaturia has not been investigated. It might be by activating GRSK, phosphorylation of NHERF1 Ser 290, and inactivation of NaPi-2a, as shown for FGF23 (Section 6.3 and Section 6.4), with heparan sulfate substituting for klotho as a co-receptor.

## 12. Clinical Applications of the Expanding Knowledge of Phosphate Transport

Already, knowledge of the cellular mechanisms has indicated potentially useful ways to treat conditions with disordered phosphate metabolism, summarized in Table 4. Treatment with Burosomab, an FGF23 antibody, is now an approved therapy for X-linked hypophosphatemic rickets and inoperable TIO, to tackle persistent hypophosphatemia due to a primary increase in FGF23 production [32]. In cystic fibrosis due to F508del CFTR, the CFTR corrector VX-809 increases the binding affinity between NHERF1 PDZ1 and the mutant protein and its membrane stability [358,359]. Perhaps small molecules with a similar action may have a place in controlling hypophosphatemia. In chronic renal failure (CRF), very high levels of FGF23 are secondary to phosphate retention and probably contribute to increased cardiovascular morbidity and mortality [31]. Here, various interventions are being explored to reduce plasma phosphate in advanced CRF. Tenapanor, an agent which inhibits NHE3 and reduces paracellular phosphate absorption from the intestine [7], now has Food and Drug Administration US (FDA) approval for use in patients with end-stage renal failure receiving dialysis. The calcimimetic cinacalcet which binds to the CaSR and decreases PTH secretion reduces phosphate concentrations indirectly. Another approach under investigation is to increase FGF23 clearance by inhibiting O-glycosylation by GALNT3, which normally protects FGF23 from proteolysis. Use of FGF23 C-terminal fragments to block interaction of FGF23 with FGFR1c-klotho and of other agents to inhibit FGFR signalling are also being considered [32]. The demonstration that PTH signalling continues after PTH/PTH1R internalisation [23] is driving research to develop a PTHrP derivative with extended activity which will enhance the benefits of intermittent treatment with abaloparatide, a human parathyroid hormone-related peptide analogue, on bone mineral density in osteoporosis [9,23,112]. An important, but less obvious, benefit of the molecular findings is that they will guide selection of genes and pathways for interrogation in the genomic data generated by on-going GWASs.

## 13. Summary and Conclusions

Years of research have established the physiology of phosphate regulation. The introduction and availability of a bewildering array of new technologies have enabled detailed exploration of the cellular mechanisms that mediate this at a molecular level. The findings are revealing an amazing, intricately regulated system within individual cells which co-ordinates with neighbouring cellular activity. Some findings which particularly merit highlighting are the following: (1) the incredible versality of NHERF1 (see below), (2) the essential functions of inherently disordered protein sequences in regulating interaction of distant PDZ domains with their PDZ-binding partners, (3) the increasing evidence of the promiscuity of the PDZ/PDZ-binding partnerships, (4) the prolonged activity of PTH after endocytic uptake of the liganded PTH1R receptor, and (5) the discovery that the primate isoform of RGS14 binds to NHERF1 and inhibits phosphate transport.

How does NHERF1 coordinate the simultaneous activities of a multiplicity of membrane-bound proteins? One NHERF1 PDZ domain can only bind one protein at a time, and even a raft of NHERF1 proteins can only accommodate a handful of proteins. Factors which may contribute to the explanation are, first, that there appears to be a large surfeit of NHERF1 relative to the amounts of competing proteins competing for binding; hence, many PDZ domains may be unoccupied. Second, interactions with NHERF1 are transitory and very brief. Third, selection of a binding ligand for the PDZ domains depends on interaction of amino acid side chains in and around the NHERF1 PDZ domains with those of residues upstream of the ligand C-terminal [93]. Allosteric conformation induced by phosphorylation/ dephosphorylation of NHERF1 [11] will change the orientation of side chains of amino acids in and around the PDZ domains. This will change the selection of protein ligand bound at the PDZ domain.

Inevitably as our vision of the cellular processes involved in phosphate absorption expands, so does the list of questions/issues to resolve. Amongst them are the following:(1)Whether interaction of full-length PTH, the PTH1R receptor, and G-signalling are the same as for the PTH1–34 fragment.(2)Clearer definition of the signalling pathway via cAMP and PKA to NHERF1. It appears speculative at present.(3)Clearer definition of the signalling pathway of IGFR which increases renal phosphate absorption.(4)The form of PTH which normally activates proximal tubule apical PTH1R. Activation by filtered intact PTH or N-terminal PTH fragments seems an imprecise regulatory mechanism for such a finely controlled reabsorption system. How far is locally produced PTHrP involved?(5)The function of PTHrP in the proximal tubules postnatally.(6)How PTH signalling at the apical and basolateral membranes in the proximal tubule are co-ordinated.(7)The location of NHERF1 at the BBM. Logically, it should be in the (long) cilia close to apically sited NaPi-2a, but findings are conflicting.(8)The roles of RAMPS in PTH/PTH1R signalling.(9)The function of the internal PDZ-binding motif of NaPi-2a.(10)Whether RGS14 has a role in regulating phosphate transport through inactivation of PTH/PTH1R signalling and/or in humans through blocking NaPi-2a inactivation by PTH.(11)Whether Epac is stimulated in parallel with PKA by PTH/PTH1R. If it is, do PKA and Epac operate an activation/inhibitory partnership to regulate PTH activity?(12)The dysfunctional protein activity which is being highlighted in GWASs of calcium stone formers.(13)Whether MEPE has a physiological role in the kidneys.

The on-going research into the molecular mechanisms of renal phosphate absorption is revealing an amazingly intricate system which has evolved over millions of years. The emerging findings are providing greater insight into failures of the process and indicating new avenues for therapeutic intervention. Parallel studies are similarly generating a wealth of new information about the mechanisms of phosphate turnover in bone. Ultimately, these must be married to the renal findings to obtain a more complete picture to optimise therapy.

## Figures and Tables

**Figure 1 ijms-25-04684-f001:**
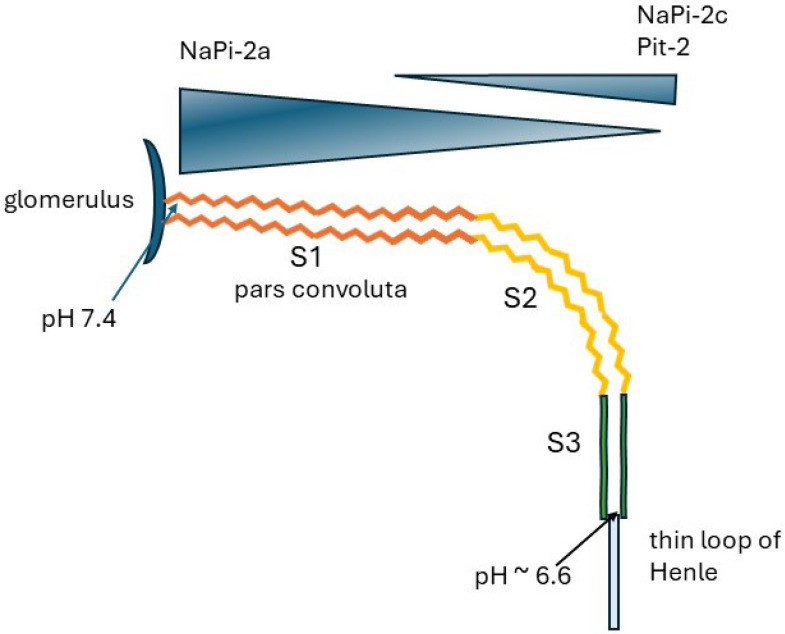
Schematic of the proximal renal tubule, not drawn to scale, showing the three segments S1 (pars convoluta), S2, and S3 (pars recta). The sodium phosphate 2a (NaPi-2a) is the dominant phosphate transporter with highest concentration in the first part of S1. Activity of sodium phosphate 2c (NaPi-2c) and the sodium-dependent transporter, Pit-2 [PiT-2], increases along the tubule and pH falls.

**Figure 2 ijms-25-04684-f002:**
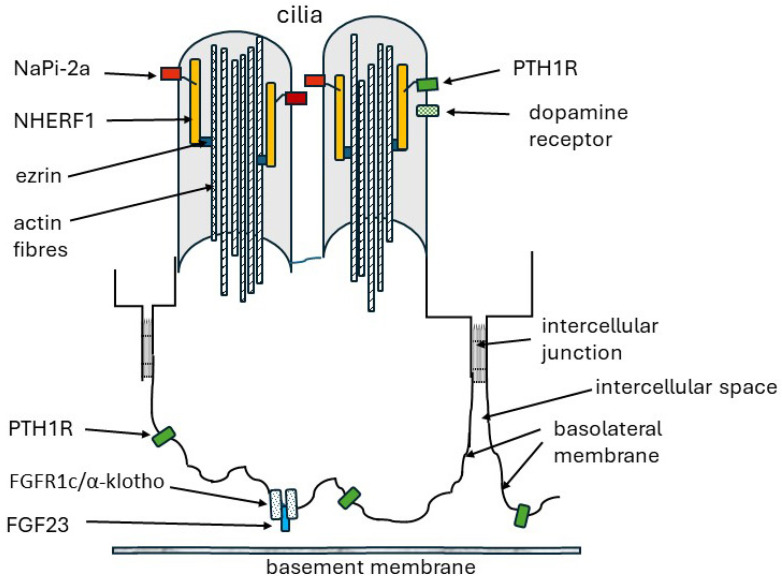
A vertical section through an epithelial cell of the proximal renal tubule. NaPi-2a in the apical brush border membrane binds with the scaffolding protein NHERF1 and is thereby tethered to the actin core of the cilia via the linking protein ezrin. Parathyroid hormone (PTH) and dopamine, which regulate NaPi-2a, bind to receptors in the cilia. PTH also binds to its receptor in the basolateral membrane, as does the other regulatory hormone FGF23. PTH1R: PTH 1 receptor; FGFR1c/α-klotho: the FGF23 receptor.

**Figure 3 ijms-25-04684-f003:**
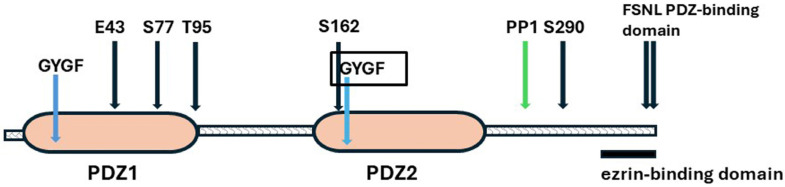
Schematic of NHERF1 based on findings of Zhang et al., 2019 [11], and Bhattacharya et al., 2019 [89]. E43: essential residue for NaPi-2a binding; S77, T95, S162, S290: key phosphorylation sites for PTH inactivation of Na-Pi-2a; PP1: protein phosphatase 1; GYGF: core PDZ binding motif; FSNL: C-terminal PDZ-binding domain.

**Figure 4 ijms-25-04684-f004:**
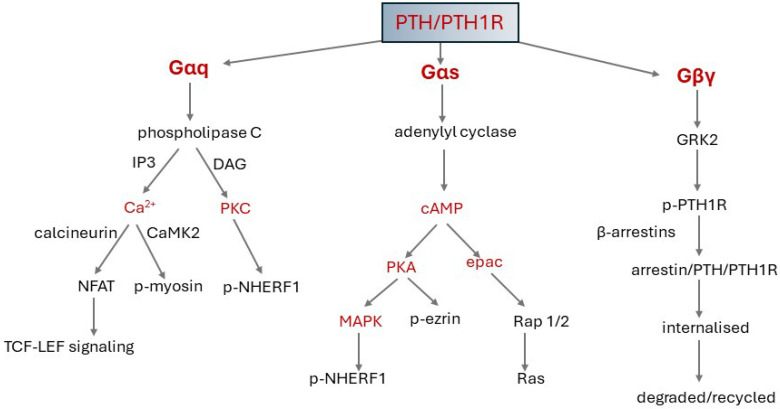
G-protein signalling by PTH activation of the PTH1R receptor. GRK2 G protein-coupled receptor kinase 2, IP3 inositol trisphosphate, DAG diacylglycerol, PKC protein kinase C, CAMKII Ca^2+^/calmodulin-dependent protein kinase II, NFAT calcineurin/nuclear factor of activated T cells signalling, TCF-LEF T-cell factor/lymphoid enhancer binding factor transcription factor, PKA protein kinase A, epac exchange protein directly activated by c-AMP, Rap 1/2 Ras-associated protein 1/2, Ras GTPase, PTH1R parathyroid hormone 1 receptor.

**Figure 5 ijms-25-04684-f005:**
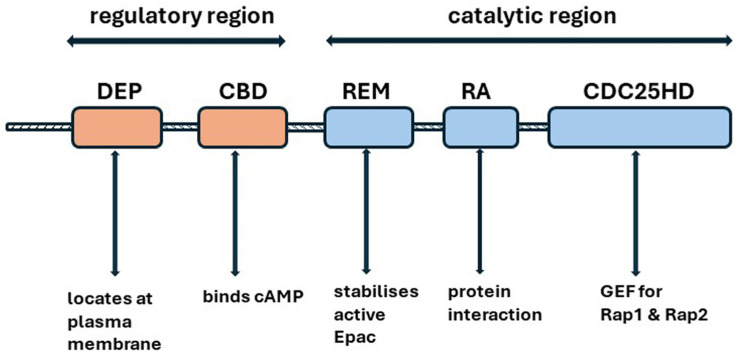
Schematic of Epac1 (exchange protein directly activated by c-AMP 1). DEP dishevelled/Egl-10/pleckstrin domain, CBD cAMP-binding domain, REM RAS exchange motif domain, RA RAS association domain, CDC25HD cell division cycle 25-homology domain.

**Figure 6 ijms-25-04684-f006:**
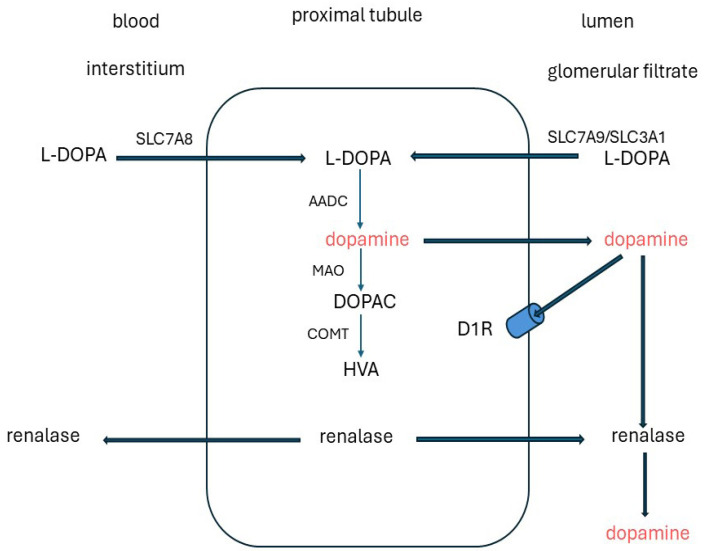
Dopamine production and metabolism in the proximal renal tubule. l-DOPA 3,4-dihydroxy-l-phenylalanine, DOPAC 3,4-Dihydroxyphenylacetic acid, HVA Homovanillic acid, D1R dopamine 1 receptor, SLC7A8 and SLC7A9/SLC3A1 amino acid transporters, AADC aromatic L-amino acid decarboxylase, MAO monoamine oxidase, COMT catechol-*O*-methyltransferase.

**Figure 7 ijms-25-04684-f007:**
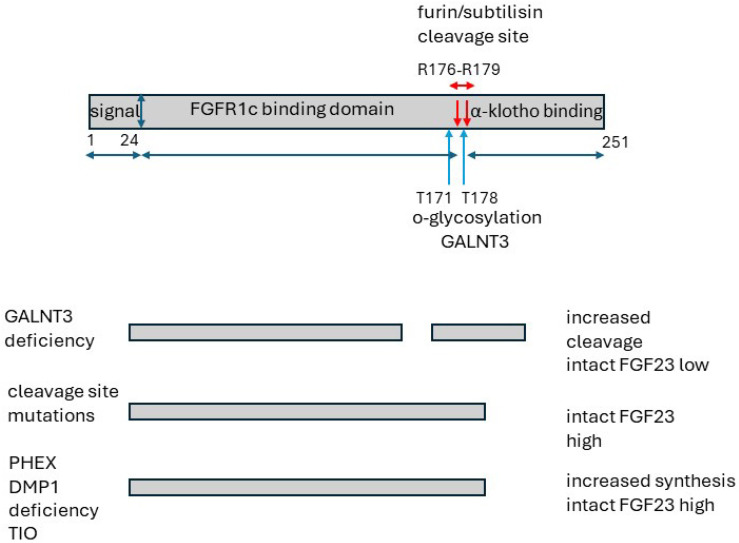
The structure of FGF23 showing the signalling peptide and the furin/subtilisin cleavage site RXXR at R176–R179 where the C-terminal α-klotho binding domain is detached from intact FGF23 intracellularly. Enzyme cleavage is prevented by O-glycosylation of Thr178 by GALNT3 (N-acetylgalactosaminyl-transferase 3), promoted by preliminary Thr171 glycosylation. PHEX: phosphate regulating endopeptidase homolog X-linked; DMP1 dentin matrix protein 1; TIO tumour-induced osteomalacia.

**Figure 8 ijms-25-04684-f008:**
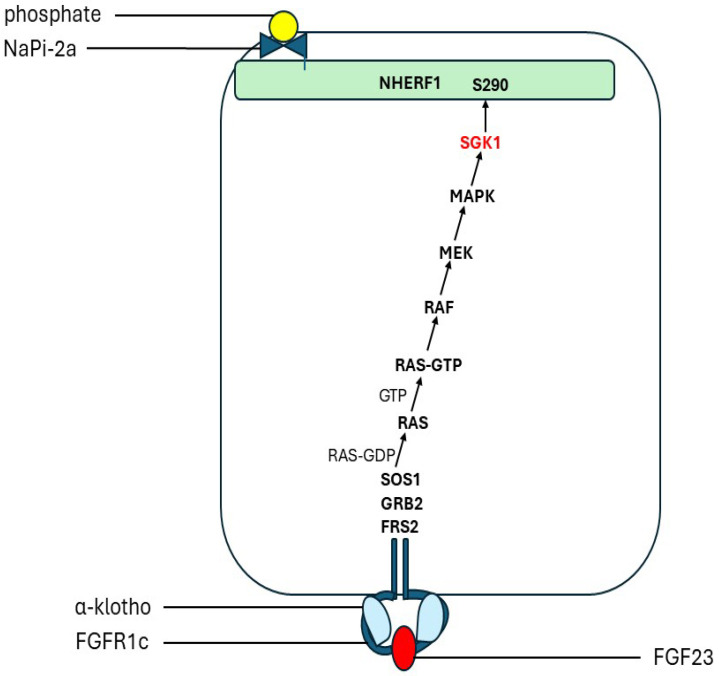
FGF23/FGFR1c-αklotho signalling pathway which mediates phosphorylation of NHERF1. FGF23 decreases phosphate absorption through activation of the ERK1/2-SGK1 signalling pathway and phosphorylation of NHERF1 [237,242,248]. As for PTH, phosphorylation of Ser 290 is proposed as a central event [11,93]. FRS2 Fibroblast growth factor (FGF) receptor substrate 2, GRB2 Growth factor receptor-bound protein 2, SOS1 Son of sevenless homolog 1 (SOS Ras/Rac Guanine Nucleotide Exchange Factor 1), SGK1 Serum/Glucocorticoid-Regulated Kinase 1.

**Figure 9 ijms-25-04684-f009:**
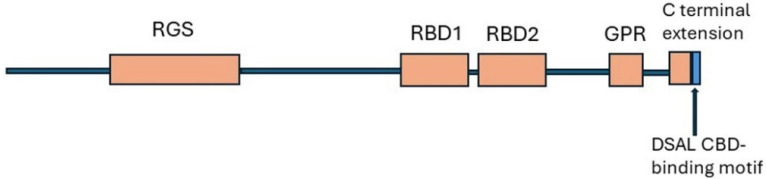
Schematic of human Regulator of G protein signalling 14 [RGS14]. RGS Regulator of G protein signalling motif, RBD1/RBD2 Ras/Rap-binding domains 1 and 2, GPR G-protein regulator (alias GoLoco) motif, CBD DSAL C-terminal PDZ-binding motif. The C-terminal RGS14 extension found in humans and primates is indicated.

**Figure 10 ijms-25-04684-f010:**
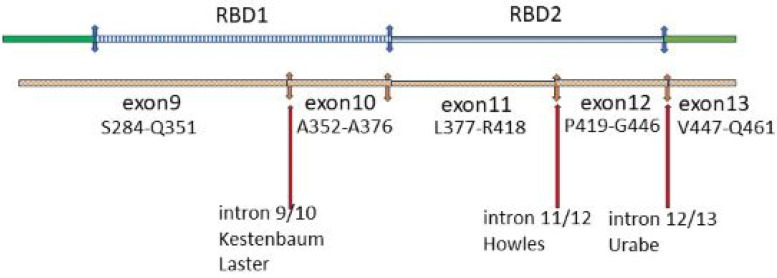
Single nucleotide polymorphisms (SNPs) in the RAP 1 and 2 binding domain of RGS14 identified in genome-wide association studies for kidney stones or plasma phosphate. Studies: Kestenbaum et al., 2010 [275], Laster et al., 2022 [276], Howles et al., 2019 [271], Urabe et al., 2012 [272], Tanikawa et al., 2019 [274]. A fourth SNP was identified up-stream, in the RGS domain of GS14 [270,273]. RBD1 and RBD2 Ras/Rap-binding domains 1 and 2.

**Figure 11 ijms-25-04684-f011:**
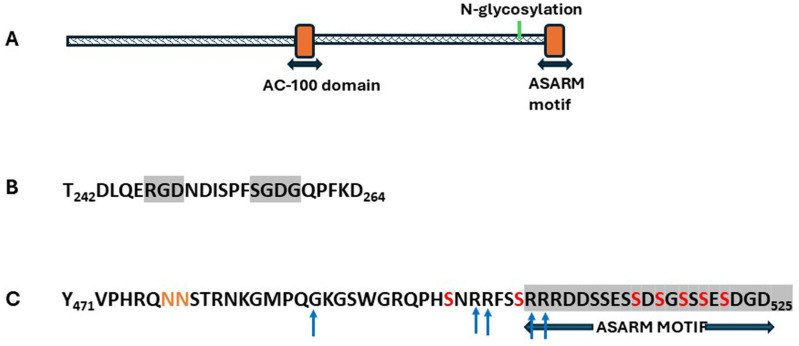
Structure of MEPE (matrix extracellular phosphoglycoprotein), based on PSN [327]. (**A**): Schematic of intact MEPE showing locations of the AC-100 domain, ASARM motif, and N-glycosylation site. (**B**): Amino acid sequence of the AC-100domain. RGD integrin binding motif, SGDG glycosaminoglycans-binding motif. (**C**): Proteolytic cleavage sites (arrows). NN N-glycosylation sites, S (red) serine residues, five are casein kinase (FAM2C) phosphorylation sites.

**Table 1 ijms-25-04684-t001:** Inherited disorders of PTH/PTHR1 signalling.

Disorder	Gene	Protein	Plasma: Ca^2+^	Pi	PTH	FEPO4	Urine: cAMP Post PTH	Bone Deformity
Jansen’s Metaphyseal chondrodysplasia [9,149,150,151]	*PTH1R*	PTH/PTHrP receptor	↑	↓	low/ND	↑	-	Yes
Isolated PTH deficiency [146,152]	*PTH*	PTH	↓	↑	low/ND	↓	-	No
^†^ Pseudohypo-parathyroidism 1A [9,145,147,153,154]	*GNAS* from mother	Gsα	↓	↑	↑	↓	↓	Yes
^†^ Pseudo-pseudo hypoparathyroidism [9,145,147,153,154]	*GNAS* from father	Gsα	norm	norm	norm	norm	norm	Yes

^†^*Gsα* is an imprinted gene. In most tissues, including bone, there is biallelic expression. In the renal cortex, *Gsα* expression from the paternal allele is silenced. A Gsα mutation inherited from mother causes disturbances in bone and the renal cortex. The same mutation inherited from father has damaging effects on bone, but without a renal disturbance. There are also pseudohypoparathyroidism subclasses 1B and 1C with decreased renal PTH responses due to imprinting abnormalities, reviewed in [9,145,147]. FEPO4: Fractional excretion of phosphate, norm-normal concentrations, ND: not detectable. ↑ increased, ↓ decreased.

**Table 2 ijms-25-04684-t002:** Genetic disorders associated with abnormal circulating concentrations of intact FGF23 ^†^.

Disorder	Gene Defect	Mechanism	Clinical Manifestations
Increased intact FGF23			
Familial X-linked hypophosphatemic rickets (XLH) OMIM 193100	*PHEX* deficiency.	Increased FGF23 synthesis.	Hypophosphatemic rickets; dental abnormalities.
Autosomal recessive hypophosphatemia OMIM 241520	*DMP1* deficiency.	Increased FGF23 synthesis.	Hypophosphatemic rickets; dental abnormalities.
Tumor-induced osteomalacia (TIO)	*FN1-FGFR1* fusion gene.	Increased FGF3 synthesis in tumours.	Hypophosphatemia, muscle weakness, osteomalacia, fractures.
Linear nevus sebaceous syndrome OMIM 163200	Sporadic *HRAS*, *KRAS*, and *NRAS* deficiencies.	Increased FGF 23 synthesis possibly by the skin nevi--etc? By the skin nevi-similar to TIO.	Linear sebaceous nevi (face), central nervous system, ocular abnormalities and skeletal defects.
McCune-Albright syndrome (fibrous dysplasia)	Somatic *GNAS* activating mutations of GSα subunit.	Increased FGF23 synthesis in fibrous bone [228,229].	Bone and endocrine abnormalities, skin pigmentation, 50% have raised FGF23 and hypophosphatemia.
Osteoglophonic dysplasia	*FGFR1c*-activating mutation.	Increased FGF23 synthesis possibly in bone-? in bone [19].	Craniosynostosis, dwarfism, non-ossifying bone lesions, some have hypophosphatemia.
Non-lethal Raine syndrome	*FAM20C* deficiency—a kinase that normally phosphorylates FGF23 S180 that promotes cleavage.	Decreased FGF23 cleavage [230,231] by furin/subtilisin and increased FGF23 synthesis in bone.	Typically fatal neonatally, bone hyper-density, lung hypoplasia. Survivors have raised FGF23 and hypophosphatemia.
Autosomal dominant hypophosphatemic rickets (ADHR)	*FGF23* cleavage site mutations.	No cleavage of intact FGF23 by furin/subtilisin pre-secretion.	Presentation in children: hypophosphatemic rickets, dental problems; in adolescents/adults: bone pain, weakness, pseudofractures.
Decreased intact FGF23			
Familial tumoral calcinosis	*GALNT3* deficiency.	Loss of glycosyl protection at furin/subtilisin cleavage site, leads to excessive FGF23 degradation.	Low intact FGF23, raised phosphate and 1,25[OH]_2_D. Periarticular, vascular subcutaneous calcium phosphate deposits. Painful dense bones.

^†^ Refer to [9,16,18,19,20] for reviews. Genes: *PHEX phosphate regulating endopeptidase homolog X-linked; DMP1 dentin matrix protein 1; FN1-FGFR1 fusion gene fused fibronectin 1 and fibroblast growth factor receptor 1 genes; HRAS, KRAS, and NRAS GTPases from the RAS oncogene family; GNAS Adenylate Cyclase-Stimulating G Alpha Protein; FGFR1c fibroblast growth factor receptor 1c; FAM20C: family with sequence similarity 20, member C (Golgi Casein Kinase); GALNT3 N-acetylgalactosaminyl-transferase 3*.

**Table 3 ijms-25-04684-t003:** Studies to investigate MEPE and ASARM function in kidneys.

Study	Procedure	Relevant Findings
**MEPE infusion**		
Rowe et al., 2004 [323]	Wt mice; boluses of human MEPE for 31 h	↓ serum phosphate, ↓ 1,25[OH]_2_D ↑ FEPO4
Dobbie et al., 2008 [331]	Wt rats; saline or MEPE infused at three doses for 2 h GFR measured with inulin.	Dose-dependent ↑ FEPO4
Shirley et al., 2010 [332]	Wt rats; 2 h infusion of vehicle or MEPE; fluid from proximal renal tubules	Significant ↑ FEPO4
**Animal Models**		
Gowen et al., 2002 [333]	Mouse *mepe* KO at 4 m and 12 m	Homoz^−/−^, Heteroz^−/+^: bone mass significant ↑ serum phosphate as wt
Zelenchuk et al., 2015 [334]	*Group 1*: Mouse *mepe* KO up to 22 m	at 22m ↑ serum phosphate; ↑ FEPO4 Kidney mRNA v. wt: ↑ *NaPi-2a*; ↑ *NaPi-2c* ↑ Bone mass
David et al., 2009 [330] ^†^	Transgenic mouse with over-expressed *mepe* up to 19 weeks normal phosphate and vit D intakes	MEPE protein expressed in bone and kidneys ↑ number of renal blood vessels; ↓ diameter ↑ serum phosphate; ↓ FEPO4; ↑ renin/aldosterone Kidney mRNA ↑ *NaPi-2a*; ↓ *NaPi-2c* ↑ *VEGFR* ↑ Bone mass; Bone mineral ↓; osteoid ↑
**Renal Cell studies**		
Rowe et al., 2004 [323]	^33^PO_4_ uptake: human proximal tubule cells and renal cell line treated with MEPE for 3–4 h	significant ↓ ^33^PO_4_ uptake; ↓ Vmax Km no change
Marks et al., 2008 [335]	Rats infused with MEPE for 3 h. analyses of BBM vesicles	NaPi-2a: significant dose-dependent ↓ NaPi-2c: trend for an ↑
**ASARM infusion**		
David et al., 2011 [336]	Wt mice aged 8 to 12 weeks continuous infusion of ASARM or vehicle; normal phosphate diet	Serum: 2-fold ↑ ASARM and 1,25[OH]_2_D vit D; ↑ FGF23 ↓ phosphate; PTH no change. ↑ FEPO4
**Animal Model**		
Zelenchuk et al., 2015 [334] ^††^	*Group 2:* Transgenic Mouse *mepe* KO with over-expressed *ASARM* up to 22 m	↓ serum phosphate; ↓ 1,25[OH]_2_D vit D; ↑ PTH; ↑ FGF23; ↑ FEPO4; Kidney mRNA ↑ *NaPi-2a*; ↑ *NaPi-2c*; ↓ bone mineral

^†^ Marked renal vasculature changes affecting Na^+^ balance. ^††^ Significantly reduced body weight (approx. 80%). Raised FGF23 would contribute to increased phosphate excretion and low 1,25[OH]_2_D. FEPO4 Fractional excretion of phosphate; BBM brush border membrane VEGFR vascular endothelial growth factor receptor. ↑ increased; ↓ decreased.

**Table 4 ijms-25-04684-t004:** Novel approved or potentially useful therapies for phosphate management.

Clinical Aim	Therapeutic Aim	Therapeutic Approach	Therapeutic Agent	Status
Reduce cardiovascular risk from high FGF23 in chronic renal failure (CRF)	Decrease excessive bone production of FGF23 in CRF driven by high phosphate.	Reduce intestinal absorption of phosphate	Tenapanor: inhibits NHE3-(Na^+^/H^+^ ion exchanger) [7]	Approved for end-stage renal failure
		Calcimimetics to decrease PTH, bone turnover and FGF23 (secondary unexplained effect)	Cinacalcet Etacalcetide [31,360]	In routine use to decrease PTH in CRF
	Reduce calcium phosphate deposition	Infusion of the MEPE ASARM peptide	ASARM peptide [28]	Experimental. Reduces vascular calcification in Mice with CRF
Correct hypophosphaturia and abnormal bone formation in genetic disorders and TIO with a primary increase in FGF23	Decrease circulating intact FGF23	Prevent thr 178 glycosylation of intact FGF23, thereby facilitating proteolytic cleavage of FGF23 by furin	T3 Inh-1 a small molecule inhibitor of GALNT3 [361]	Experimental. Effective in normal mice
	Decrease the renal response to FGF23	Block binding of FGF23 to the FGFR1c-αklotho receptor	Burosumab: a recombinant human IgG monoclonal antibody [32]	Approved for X-linked hypophosphatemic rickets and inoperable TIO
		Block binding of FGF23 to the FGFR1c-αklotho receptor with C-terminal FGF23 fragments which do not activate signalling	FGF23-C-terminal fragments [29]	Experimental. Effective in cell cultures and Hyp mice
		Inhibit FGFR1c-αklotho /FGF23 signalling	No specific small molecule inhibitors available	Still speculative -clinical safety and merits questionable
Prolong the action of PTH for treatment of hypoparathyroidism	Prolong cAMP signalling response to PTH	Use a long-acting PTH hybrid to stabilise binding to the PTH1R receptor. in G-protein -independent receptor conformation (R^0^)	Long-acting PTH ligands [PTH/PTHrP hybrid peptides] [34]	Experimental
Reduce effects of increased PTH in primary hyperparathyroidism or PTHrP in hypercalcaemia of malignancy	Reduce activity of the PTH1R receptor	Reduce the PTH1R response by binding with a small molecule modulator	A small modulator, Pitt12, is being optimised [23]	Experimental
Improve the efficacy of intermittent treatment with PTHrP to promote bone anabolism in post-menopausal osteoporosis	Prolong the duration of action of PTHrP in bone	Develop analogues to localise in endosomes and generate endosomal cAMP to ensure nuclear delivery	Under investigation [23]	Experimental
Increase phosphate reabsorption in phosphate wasting disorders	Increase expression of NaPi-2a in the brush border membrane	Increase the binding affinity between NHERF1 PDZ1 and NaPi-2a and increase its membrane stability	The CFTR corrector VX-809 used to treat cystic fibrosis due to the CFTR F508del increases surface expression and activity of mutant CFTR [358,359]	This is a speculative approach which has not been explored for NaPi-2a

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
