# Peer review of "The Intricacies of Renal Phosphate Reabsorption—An Overview"

_ijms, 2024, doi:10.3390/ijms25094684_

Round 1
Reviewer 1 Report
Comments and Suggestions for Authors
Valerie Walker summarizes the molecular mechanisms of renal phosphate reabsorption in her descriptive review. The text is generally well-written and thoroughly discusses the hormonal and molecular mechanisms of phosphate homeostasis using a lot of references. The topic is interesting and has many novelties. Writing this detailed manuscript alone must have been a huge effort. I have only minor remarks and suggestions:
1. Adding a Figure that summarizes the cellular localization and main function of renal phosphate transporters and NHERF1 (to lines 82-249) would be helpful.
2. Fig. 2. Panel A contains only text. Please draw the mentioned mechanisms or delete this part of the Figure.
3. Please add a Table or Figure that summarizes the used drugs and promising candidate molecules with mode of action on phosphate reabsorption.
Author Response
Thank you for making time to work through my long review and for your helpful comments and suggestions with which I concur entirely
1.Adding a Figure that summarizes the cellular localization and main function of renal phosphate transporters and NHERF1 (to lines 82-249) would be helpful.
I was a little uncertain whether anatomical or intracellular localisation was required and decided that probably both were needed. Hence I have drawn two new figures. Fig.1 shows localisation in the proximal tubule: starts line 105 Fig.2 shows localisation in a proximal tubular cell. To match the text I have put it later: starts line 221 I have re-numbered the remaining figures
2. Fig. 2. Panel A contains only text. Please draw the mentioned mechanisms or delete this part of the Figure.
I deleted Panel 2A and put the content into the text: 356-360 I deleted B from panel 2B, which is now Fig. 4
3. Please add a Table or Figure that summarizes the used drugs and promising candidate molecules with mode of action on phosphate reabsorption.
I have added Table 4: starts line 1079 and three new references 363-365
Reviewer 2 Report
Comments and Suggestions for Authors
The paper "The Intricacies of Renal Phosphate Reabsorption-An Overview" submitted for review consists of an exhaustive review, the result of a very thorough and precision work, which attempts to give an interpretation of Phosphate metabolism through the analysis of the cellular and biochemical mechanisms of the various components of the system.
The review examines in great detail the various factors involved in maintaining phosphate homeostasis in the body, and any genetic or metabolic errors conditioning important pathologies.
The picture shown is complex and not easy to read for non-experts (especially reference to clinical practitioners).
Exhaustive bibliography.
The work deserves to be published with some minor observations:
1. Make the diagrams present easier to read, including graphically
2. Throughout the text there is multiple use of acronyms and abbreviations that would deserve a paragraph of their own to improve reading
3. On lines 871and 872 there is an unspecified abbreviation "TIA" when the paragraph talks about "MEPE" in the chapter on tumor-induced osteomalacia (TIO)
4. Probably less extension, even at the risk of losing some detail, would improve the reading.
5. In the conclusion, open questions from the cellular and biochemical point of view are emphasized more than an overall view of the several described steps.
Author Response
Thank you for making time to go through my long review and for your helpful comments and suggestions with which I concur entirely.
- Make the diagrams present easier to read, including graphically
I have tried to make the graphical abstract clearer- see revised figure.
With the other figures, all I can do at this stage is to make the legends clearer by expanding the abbreviations (done). Should the review be accepted, hopefully the publishers will be able to improve on my layout.
- Throughout the text there is multiple use of acronyms and abbreviations that would deserve a paragraph of their own to improve reading
Because there are so many, I have compromised: All abbreviations used in the Figs are expanded in the legends; Acronyms of importance or which recur frequently are included in an Abbreviations list at the end of the script.The remainder are expanded in the text when first mentioned.
- On lines 871and 872 there is an unspecified abbreviation "TIA" when the paragraph talks about "MEPE" in the chapter on tumor-induced osteomalacia (TIO)
Thank you. TIA was wrong. I have changed to TIO. (now located to lines 918 & 919)
- Probably less extension, even at the risk of losing some detail, would improve the reading.
I agree that it is not very digestible, but I wanted to produce an accurate summary & not cut corners at this stage. This is partly because I was very aware that I am summarising other people’s work and have a duty to do this carefully; also, because the picture is evolving and our ideas are changing, it is important that we work on accurate information.
- In the conclusion, open questions from the cellular and biochemical point of view are emphasized more than an overall view of the several described steps.
I am reluctant to extend the manuscript further! However, I have added one sentence which I think is important. Line 1137.
Reviewer 3 Report
Comments and Suggestions for Authors
Dear Author:
I read your review articles on the complexity of renal phosphate absorption with interest. Thank you for trying to compile such a complicated issue in this review.
1. Because the information is quite complex, I proposed that the author consider having abbreviation library so the reader can follow the information easier.
2. Also, there are many abbreviations that present without appropriate full words. All abbreviations should follow the full words.
3. All figures should be enlarged with appropriate legends (The graphic abstract is so small and not clear what the author point.
4. Legends in figure may not need to provide detail of references, but more importantly is to spell out all abbreviations, otherwise it may not be clear for the reader.
5. Please verify if NaPi-2a is NaPt-2a or it is typo?
6. Table 1: it should probably have the lines between cells to make it easier to read.
7. Table 2: It would be nice to summarize each disorder, features of the diagnosis and mechanism.
8. It would be useful if the author can provide the cartoon showing the synthesis of FGF23, secretion, cleavage, and what regulation of FGF23 related with the disorders of PHEX, DMP1, GALNT3 genes.
Thank you for your hard work.
Reviewer
Author Response
Comments and Suggestions for Authors Thank you for making time to go through my long review and for your helpful comments and suggestions with which I concur entirely.
Dear Author:
I read your review articles on the complexity of renal phosphate absorption with interest. Thank you for trying to compile such a complicated issue in this review.
- Because the information is quite complex, I proposed that the author consider having abbreviation library so the reader can follow the information easier.
- Also, there are many abbreviations that present without appropriate full words. All abbreviations should follow the full words.
These comments are linked and the issue of heavy use of acronyms was also raised by another of the Reviewers. Because there are so many, I have compromised: All abbreviations used in the Figs are expanded in the legends; Acronyms of importance or which recur frequently are included in an Abbreviations list at the end of the text. The remainder are expanded in the text when first mentioned.
- All figures should be enlarged with appropriate legends (The graphic abstract is so small and not clear what the author point.
I have tried to make the graphical abstract clearer- see revised figure.
With the other figures, all I can do at this stage is to make the legends clearer by expanding the abbreviations (done). Should the review be accepted, hopefully the publishers will be able to enlarge the figures and improve on my layout.
- Legends in figure may not need to provide detail of references, but more importantly is to spell out all abbreviations, otherwise it may not be clear for the reader.
I have expanded all abbreviations in the Fig. legends
I have included references only when I have used the experimental findings of particular studies to design my figures.
- Please verify if NaPi-2a is NaPt-2a or it is typo?
Thank you. NaPt-2a is wrong. I have corrected it on Lines 273, 274, ad in Table 3
- Table 1: it should probably have the lines between cells to make it easier to read.
I have done this-see line 447
- Table 2: It would be nice to summarize each disorder, features of the diagnosis and mechanism.
I have added information (Table 2 =-now starts on line 614), but since this is not a clinical paper have directed readers to good clinical reviews for more information.
- It would be useful if the author can provide the cartoon showing the synthesis of FGF23, secretion, cleavage, and what regulation of FGF23 related with the disorders of PHEX, DMP1, GALNT3 genes.
I have drawn a new Fig (Fig 7) -line 597 following section 6.2
This led me to change the order of this section to a more logical sequence:
I have moved the original lines 571-577, starting with ‘Individuals with inherited disorders—’ to line 605 after the figure, and Table 2 to line 614 to follow on from this.